# Primosomal protein PriC rescues replication initiation stress by bypassing the DnaA-DnaB interaction step for DnaB helicase loading at *oriC*

**Ryusei Yoshida, Kazuma Korogi[†], Qinfei Wu[†], Shogo Ozaki\*, Tsutomu Katayama\***

Department of Molecular Biology, Kyushu University Graduate School of Pharmaceutical Sciences, Fukuoka, Japan

**\*For correspondence:**
shogo.ozaki@phar.kyushu-u.ac.jp (SO);
katayama@phar.kyushu-u.ac.jp (TK)

[†]These authors contributed equally to this work

**Competing interest:** The authors declare that no competing interests exist.

## eLife Assessment

This manuscript reports findings of **fundamental** significance on how bacteria might load helicase for DNA replication when normal DnaA-based loading pathway is defective. It provides **convincing** genetic and biochemical evidence that helicase loading at the *E. coli oriC* is not (as previously assumed) exclusively performed by the DnaA initiator protein but can also be executed by PriC (whether this occurs specifically at oriC has not been addressed in vivo). This is a significant step forward in our understanding of bacterial replication initiation.

**Abstract** In *Escherichia coli*, replisome and replication fork assembly is initiated by DnaB helicase loading at the chromosomal origin *oriC* via its interactions with the DnaA initiator and the DnaC helicase loader. Upon replication fork arrest, the replisome including DnaB dissociates from the stalled fork. Replication fork progression is rescued by primosomal protein PriA- or PriC-dependent pathway in which PriA and PriC promote reloading of DnaB in different mechanisms. However, the mechanism responsible for rescue of blocked replication initiation at *oriC* remains unclear. Here, we found that PriC rescued blocked replication initiation in cells expressing an initiation-specific DnaC mutant, in mutant cells defective in DnaA-DnaB interactions, and in cells containing truncated *oriC* sequence variants. PriC rescued DnaB loading at *oriC* even in the absence of Rep helicase, a stimulator of the PriC-dependent replication fork restart pathway. These results of in vitro reconstituted assays concordantly suggest that this initiation-specific rescue mechanism provides a bypass of the DnaA-DnaB interaction for DnaB loading by PriC-promoted loading of DnaB to the unwound *oriC* region. These findings expand understanding of mechanisms sustaining the robustness of replication initiation and specific roles for PriC in the genome maintenance.

## Introduction

Chromosome replication is strictly regulated to ensure successful duplication of the genetic material (*Costa and Diffley, 2022*; *Grimwade and Leonard, 2021*; *Kasho et al., 2023*). In *Escherichia coli*, the ATP-bound form of the replication initiator protein DnaA (ATP-DnaA) forms a higher-order complex with DNA-bending protein IHF at the unique chromosomal origin *oriC*. This complex promotes local unwinding of *oriC* and recruits a pair of DnaB helicases, which are successively loaded to the single-stranded (ss) DNA regions in a bidirectional manner with the aid of DnaC helicase loader. ssDNA-loaded DnaB interacts with DnaG primase and DNA polymerase III holoenzyme, resulting in assembly of the replisome (*Arias-Palomo et al., 2019*; *Chodavarapu and Kaguni, 2016*; *Hayashi et al., 2020*;

*O'Donnell et al., 2013*; *Sakiyama et al., 2017*; *Sakiyama et al., 2018*; *Sakiyama et al., 2022*; *Wegrzyn and Konieczny, 2023*). Replisomes disassemble when the replication fork stalls in front of an obstacle, such as protein-DNA roadblock or single-strand breaks. PriA recognizes the abandoned fork structure, which triggers reloading of DnaB with its partner proteins, such as PriB, PriC, DnaT, and DnaC (*Heller and Marians, 2005a*; *Masai et al., 1994*; *Michel and Sandler, 2017*; *Windgassen et al., 2018*). In addition, independently of PriA, PriC binds to the abandoned fork structure, triggering reloading of DnaB with the aid of DnaC, which is supported by Rep helicase depending on the fork structure (*Heller and Marians, 2005b*; *Sandler et al., 1999*). Also, homologous recombination mechanisms can rescue the abandoned replication fork during SOS responses (*Asai and Kogoma, 1994*; *Kogoma, 1997*; *Michel et al., 2001*). In contrast to these well-characterized mechanisms at abandoned forks, little is known about what happens when replication initiation is impeded at *oriC*.

*E. coli oriC* comprises a Duplex-Unwinding Element (DUE) and a DnaA-Oligomerization Region (DOR) (*Figure 1A*; *Grimwade and Leonard, 2021*; *Kasho et al., 2023*; *Wegrzyn and Konieczny, 2023*). The DUE contains 13-mer AT-rich sequence repeats known as L-, M-, and R-DUEs. The M- and R-DUEs are essential for stable DUE unwinding with the specific sequences (TT[A/G]T(T)) used for DnaA-ssDNA binding (*Ozaki and Katayama, 2012*; *Sakiyama et al., 2017*). The L-DUE containing the TTATT sequence promotes efficient DnaB loading by expanding the unwound *oriC* region (*Sakiyama et al., 2022*). The DOR is divided into three subregions: the Left-, Middle-, and Right-DORs, which contain asymmetric 9-mer DnaA binding sequences (DnaA box) with the consensus sequence TTA[T/A]NCACA (*Figure 1A*; *Noguchi et al., 2015*; *Ozaki and Katayama, 2012*; *Rozgaja et al., 2011*; *Sakiyama et al., 2017*; *Shimizu et al., 2016*). The Left-DOR contains a cluster of unidirectionally aligned DnaA boxes, including high-affinity DnaA box R1, and low-affinity boxes R5M, $\tau$1–2, and I1-2, with an IHF-binding region between R1 and R5M boxes. The Right-DOR contains an oppositely oriented DnaA box cluster including high-affinity DnaA box R4 and low-affinity boxes I3 and C1-3. These DnaA box clusters form frameworks for the Left- and Right-DnaA subcomplexes, respectively. DnaA bound to the R2 box, which solely resides in the Middle-DOR, stabilizes these DnaA subcomplexes (*Miller et al., 2009*; *Rozgaja et al., 2011*; *Shimizu et al., 2016*). In addition, the AT-cluster (TATTAAAAGAA) region, which connects to the L-DUE, stimulates DnaB loading in the absence of the Right-DnaA subcomplex (*Sakiyama et al., 2022*).

DnaA comprises four functional domains (*Figure 1B*). Domain I binds to multiple proteins such as DnaB helicases and the DnaA-assembly stimulator DiaA (*Abe et al., 2007*; *Hayashi et al., 2020*; *Keyamura et al., 2007*; *Keyamura et al., 2009*). Domain II is a flexible linker (*Abe et al., 2007*; *Nozaki and Ogawa, 2008*). Domain III contains AAA+ (ATPases Associated with various cellular Activities) motifs, which are involved in tight ATP/ADP binding, ATP hydrolysis, and domain III-III interactions (*Duderstadt et al., 2011*; *Erzberger et al., 2006*; *Felczak and Kaguni, 2004*; *Kawakami et al., 2005*; *Ozaki et al., 2012*). The arginine finger motif (Arg285) in domain III interacts with ATP bound to domain III of adjacent DnaA protomers, stimulating DnaA complex formation in a cooperative manner (*Kawakami et al., 2005*; *Noguchi et al., 2015*). In addition, H/B motifs (Val211 and Arg245) in this domain bind to ssDNA in a sequence-specific manner (*Ozaki et al., 2008*; *Sakiyama et al., 2017*). Domain IV, which comprises the helix-turn-helix motif, recognizes the DnaA box (*Fujikawa et al., 2003*).

For replication initiation, ATP-DnaA molecules cooperatively oligomerize at *oriC* to form the Left- and Right-DnaA subcomplexes (*Figure 1C*). Formation of these complexes is stimulated by homotetrameric DiaA protein, which binds up to four DnaA molecules via domain I, including Phe46 (*Keyamura et al., 2007*; *Keyamura et al., 2009*). In concert with DNA superhelicity and thermal energy, the DUE undergoes initial unstable unwinding, which is stabilized through specific interaction between H/B motifs of domain III in the Left-DnaA subcomplex and the upper strand of M- and R-DUEs. This interaction is facilitated by sharp DNA bending by IHF, resulting in stable DUE unwinding (*Figure 1C*; *Ozaki et al., 2008*; *Sakiyama et al., 2017*). When the single-stranded region is expanded to the L-DUE, the resultant single-stranded L-DUE binds the H/B motifs within the Right-DnaA subcomplex, maximizing the efficiency of the DnaB loading process (*Sakiyama et al., 2022*).

Subsequently, two DnaB helicases are loaded onto each ssDUE strand through interactions with DnaC and DnaA (*Figure 1D*). DnaC binding to DnaB changes the closed ring structure of the DnaB hexamer to an open spiral form to allow it to encircle ssDNA (*Arias-Palomo et al., 2019*; *Nagata et al., 2020*). Each Left- and Right-DnaA subcomplex binds a DnaB-DnaC complex via high-affinity

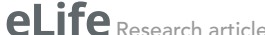

**Figure 1.** Schematic structures of *oriC* and DnaA and the replication initiation mechanism. (**A**) Overall structure of *oriC* with the AT-rich cluster. *oriC* (245 bp) includes the Duplex-Unwinding Element (DUE, purple bar) and the DnaA-Oligomerization Region (DOR, gray bar). DUE is composed of three AT-rich 13-mer repeats termed L, M, and R. Dark purple boxes indicate the specific sequences involved in DnaA-ssDNA binding (TT[A/G]T(T)). The AT-cluster (purple dots) flanking the DUE outside *oriC* is a supplementary unwinding region. The DnaA-oligomerization region (DOR) contains 12

*Figure 1 continued on next page*

*Figure 1 continued*

DnaA boxes (filled and open arrowheads representing sites with the full consensus TTA[A/T]NCACA sequence and sites with mismatches, respectively) and an IHF-binding region (IBR, green box). The DOR is subdivided into Left, Middle, and Right subregions. (**B**) Domains of DnaA. Domains I-IV are shown schematically, with amino acid residue numbers shown in each bracket. H/B motifs (V211 and R245, squares) and Arginine finger (R285, triangle) are indicated. The major functions of each domain are described on the right side of the structure. (**C**) DUE unwinding by the ssDUE recruitment mechanism. DnaA and IHF are indicated by red and green diagrams, respectively. When ATP-DnaA oligomerizes, DUE is unwound unstably by thermal motion and torsional stress. The M/R region of the DUE-upper strand (purple line) binds to R1-DnaA and R5M-DnaA via IHF-induced DNA bending, resulting in stable DUE unwinding. In addition to M/R-DUE, L-DUE is moderately unwound and interacts with the Right-DnaA subcomplex. (**D**) Mechanism for DnaB loading. Each Left-DnaA and Right-DnaA subcomplex binds to a DnaB hexamer (blue ovals) complexed with DnaC (yellow circles). (**E**) PriC-dependent fork rescue pathways. PriC (vivid red circle) promotes DnaB reloading to the abandoned fork, which exposes ssDNA regions. PriC binds to ssDNA at the replication fork, which remodels the single-stranded DNA binding protein (SSB)-ssDNA complex (for simplicity, SSB is omitted) for DnaC-dependent reloading of DnaB (*upper panel*). If the ssDNA region is accompanied with a nascent lagging strand, Rep helicase (green shape) is recruited by PriC and unwinds the lagging strand to expose the ssDNA region for DnaB loading (the lower case).

interaction between DnaA Phe46 and DnaB Leu160 and guides the DnaB-DnaC complexes to unwound ssDUE strands via a low-affinity interaction between DnaA domain III His136 and DnaB (*Figure 1B and D*). Upon interaction with ssDNA, DnaC dissociates from DnaB, which enables migration of DnaB hexamer with helicase action. The pair of DnaB helicases loaded onto the ssDNA strands progress in opposite directions, forming replisomes with DnaG and Pol III holoenzyme (*Chodavarapu and Kaguni, 2016*; *O'Donnell et al., 2013*).

Replication restart pathways ensure fork progression of the entire chromosome under conditions that trigger replisome disassembly (*Heller and Marians, 2005a*; *Lopper et al., 2007*; *Michel and Sandler, 2017*). PriA helicase-dependent pathways predominantly facilitate DnaB reloading onto abandoned forks in vivo (*Flores et al., 2002*). PriA is a 3'–5' DNA helicase and has a specific affinity for forked DNA structures comprising one parental dsDNA and two newly synthesized sister dsDNA strands. This permits PriA recognition of the abandoned replication fork and the subsequent unwinding of the nascent lagging strand (*Duckworth et al., 2023*; *Windgassen et al., 2018*; *Windgassen and Keck, 2016*). The resultant unwound ssDNA associates with PriB and DnaT to promote reloading of DnaB helicase with the aid of DnaC (*Duckworth et al., 2023*; *Heller and Marians, 2005a*; *Lopper et al., 2007*). In the absence of PriB, PriC participates in a PriA-dependent pathway (*Sandler et al., 1999*). In addition, PriA is required for *oriC*/DnaA-independent chromosomal replication called stable DNA replication (SDR) (*Masai et al., 1994*), which is promoted by UV irradiation (inducible iSDR) and loss of *rnhA* or *recG* (constitutive cSDR) (*Kogoma, 1997*). In both mechanisms, replication is initiated at triple-stranded structures comprising a dsDNA-ssDNA hybrid (D-loop) or a dsDNA-ssRNA hybrid (R-loop) generated at specific chromosomal loci. These structures mimic the abandoned fork structure, thereby allowing helicase loading to occur similarly to that during PriA-dependent fork restart (*Masai et al., 1994*).

Independently of PriA, PriC can restart replication from abandoned forks through its interaction with the ssDNA region and SSB (Single-Stranded DNA Binding protein) (*Figure 1E*; *Heller and Marians, 2005a*; *Wessel et al., 2013*; *Wessel et al., 2016*). PriC consists of N-terminal and C-terminal domains, which are composed of α helices and are connected by a short linker. The PriC C-terminal domain remodels the SSB-ssDNA complex, which is a prerequisite for the recruitment of DnaB helicase (*Heller and Marians, 2005a*; *Wessel et al., 2013*; *Wessel et al., 2016*). PriC also interacts with Rep helicase. When the length of the ssDNA gap within the abandoned fork is short, Rep helps expand the ssDNA gap, promoting PriC-mediated remodeling of the SSB-ssDNA complex (*Heller and Marians, 2005a*; *Heller and Marians, 2005b*; *Nguyen et al., 2021*). Notably, in addition to these functions at the abandoned forks, PriC is suggested to play a role in DnaA-*oriC*-dependent replication initiation under challenging conditions, as evidenced by the synthetic lethality between *priC303::kan* and a subgroup of temperature-sensitive *dnaA* mutant alleles (*dnaA46* and *dnaA508*): i.e., *priC303::kan* mutants bearing *dnaA46* or *dnaA508*, but not wild-type (WT) *dnaA*, can not grow even at 30 °C (*Hinds and Sandler, 2004*). However, the mechanisms via which PriC rescues blocked replication initiation remain elusive.

In this study, we analyzed the PriC-dependent promotion of replication initiation from *oriC* in the context of various replication initiation stresses. We found that PriC stimulated replication initiation in *dnaC2* cells, in which DnaB helicase loading at *oriC* is defective, indicating that it has a role in the rescue of blocked replication initiation. Moreover, PriC stimulated the growth of cells defective in

DnaA-DnaB interactions in addition to the growth of cells with *oriC* sequence deletions that inhibit DnaB loading. Consistent with these results, in an in vitro reconstituted system, PriC stimulated DnaB loading when the DnaA-DnaB interaction was inhibited at *oriC*. Furthermore, we found that PriC did not stimulate initiation of cSDR, demonstrating that PriC functions specifically in the DnaA-*oriC* system. Taken together, we suggest that PriC rescues blocks in replication initiation by bypassing DnaB loading at *oriC*-DnaA complexes.

## Results

### PriC stimulates DNA replication initiation in *dnaA46* and *dnaC2* cells

The temperature-sensitive *dnaA46* and *dnaA508* mutants exhibit synthetic lethality with *priC303::kan* even at 30 °C (*Hinds and Sandler, 2004*), suggesting that PriC can rescue blocks in DnaA-dependent replication initiation. However, the mechanism underlying PriC-promoted rescue of blocks in replication initiation remains a mystery. Based on previous reports demonstrating PriC binding to short oligo-ssDNA, SSB, and DnaB and its role in DnaB loading at stalled replication forks (*Wessel et al., 2013*; *Wessel et al., 2016*), we considered the possibility that PriC permits bypass of DnaA-dependent stable DUE unwinding and/or DnaB loading at the *oriC* ssDUE. To genetically test these ideas, we analyzed the impact of PriC on the growth of a temperature-sensitive *dnaC2* mutant, which is competent for ongoing replisome progression at 37–42°C, but defective in DnaB loading specifically at the *oriC* ssDUE (*Withers and Bernander, 1998*).

In spotting assays performed using serial dilutions of the cell cultures and LB agar plates, the cell growth of *dnaA46* cells was slightly inhibited at 37°C (*Figure 2A*). Consistent with the previous report, when crossed with the Δ*priC* allele, the cell growth of *dnaA46* cells but not WT *dnaA* cells was severely inhibited at 37°C (*Figure 2A*). Unlike the *dnaA46 priC303::kan* cells, the *dnaA46* Δ*priC* cells grew at 30 °C without prominent inhibition (*Figure 2A*). This difference may stem from the different strain backgrounds. Notably, the cell growth of *dnaC2* mutant at 35°C was moderately inhibited, whereas it was severely inhibited in *dnaC2* Δ*priC* double mutant (*Figure 2A*). Also, the cell growth of *dnaC2* cells and *dnaC2* Δ*priC* cells was similar at 30 °C. The observed requirement of PriC for cell growth of *dnaC2* cells at 35°C supports the idea that PriC assists in the DnaC-dependent DnaB loading step at *oriC*.

Introduction of a *priC*-encoded plasmid (pBR*priC*), but not an empty vector (pBR322), restored the cell growth of *dnaA46* Δ*priC* double mutant at 37°C (*Figure 2—figure supplement 1A, B*) and that of *dnaC2* Δ*priC* double mutant at 35°C. Also, introduction of pBR*priC* to *dnaA46* and *dnaC2* mutants with intact *priC* (*priC+*) might only slightly enhance the cell growth at these temperatures (*Figure 2—figure supplement 1A, B*), suggesting that the intrinsic level of PriC is functionally nearly sufficient.

To investigate replication initiation from *oriC*, we used flow cytometry of synchronized cells bearing *dnaA46* or *dnaC2* with or without Δ*priC*. In these experiments, cells were incubated at 42 °C or 37 °C for 80–90 min to synchronize the replication cycle at the pre-initiation step, after which incubation was continued at 30 °C for 5 min to allow initiation and then 4 hr in the presence of rifampicin and cephalexin to inhibit both replication initiation and cell division and allow replication run-out of the whole chromosome. The resultant number of chromosomes per cell, measured by flow cytometry, is known to correspond to the number of *oriC* copies per cell at the time of drug addition, an indicator of initiation activity.

A majority of the synchronized *dnaA46* and *dnaC2* cells had only a single *oriC* copy per cell after incubation at 42 °C or 37 °C, regardless of the presence of Δ*priC* (*Figure 2B and C*). In cells with intact *priC* (*priC+*), after release of initiation at 30 °C for 5 min, most cells had two *oriC* copies. However, in *dnaA46* Δ*priC* cells, only about half of the cells had two *oriC* copies after the release (*Figure 2B*), indicating moderate inhibition of initiation. In *dnaC2* Δ*priC* cells, the number of two-*oriC* cells was slightly lower than that of *dnaC2* cells (*Figure 2C*). These observations are consistent with the cell growth ability (*Figure 2A*) and with our hypothesis that PriC contributes to the rescue of blocked replication initiation by assisting DnaB loading onto the *oriC* ssDNA.

We further performed flow cytometry analysis to assess the role of PriC in replication initiation of the WT *dnaA/dnaC* cells. To estimate the number of *oriC* copies per unit cell mass (ori/mass) as a proxy for initiation activity (*Sakiyama et al., 2017*; *Sakiyama et al., 2022*), exponentially growing cells were subjected to run-out replication in the presence of rifampicin and cefalexin (*Figure 2—figure supplement 2*). When cells were grown in LB medium at 25 °C, 30 °C, or 42 °C, the ori/mass values for



**Figure 2.** Requirement of *priC* for DNA replication initiation in *dnaA46* and *dnaC2* strains. (**A**) Cell growth abilities. MG1655 and its derivatives MIT125 (MG1655 *dnaA46 tnaA*::Tn*10*), KYA018 (MG1655 *dnaC2 zjj18::cat*), KRC002 (MG1655 Δ*priC::frt-kan*), KRC004 (MG1655 *dnaA46 tnaA*::Tn*10* Δ*priC::frt-kan*), and KRC005 (MG1655 *dnaC2 zjj18::cat* Δ*priC::frt-kan*) were grown overnight. 10-fold-serial dilutions of the overnight cultures (~10⁹ cells/mL) were incubated for 16 hr at 30°C, or for 14 hr at 35/37/42°C on LB agar medium. Three independent experiments were performed. +, WT *priC*; -, Δ*priC::frt-kan*. The images of representative plates are shown. (**B**) DNA replication initiation in synchronized *dnaA46* cells with or without *priC*. Exponentially growing MIT125 (MG1655 *dnaA46 tnaA*::Tn*10*) (+) and KRC004 (MG1655 *dnaA46 tnaA*::Tn*10* Δ*priC::frt-kan*) (-) cells were synchronized to the pre-initiation stage by incubation for 90 min at 42°C in LB medium, followed by incubation for 5 min at 30°C to initiate DNA replication, which was further followed by incubation with rifampicin and cephalexin for 4 hr at 42°C to allow run-out of ongoing DNA replication. Samples were withdrawn after synchronization and run-out replication. The DNA contents of these samples were measured by flow cytometry. Representative histograms of three independent experiments are shown. For each sample, the mean number of cells with more than one chromosome equivalent after a 5 min release was quantified and is shown as a percentage with standard deviations. (**C**) DNA replication initiation in synchronized *dnaC2* cells with or without *priC*.

*Figure 2 continued on next page*

*Figure 2 continued*

Exponentially growing KYA018 (MG1655 *dnaC2 zjj18::cat*) (+) and KRC005 (MG1655 *dnaC2 zjj18::cat ΔpriC::frt-kan*) (-) cells were synchronized to the pre-initiation stage by incubation for 80 min at 37℃ in LB medium, followed by incubation for 5 min at 30℃ and further incubation with rifampicin and cephalexin as described above. Samples were withdrawn and analyzed as described above.

The online version of this article includes the following figure supplement(s) for figure 2:

**Figure supplement 1.** Role of PriC in cell growth of *dnaA46* and *dnaC2* cells.

**Figure supplement 2.** Role of PriC in cell growth of cells with wild-type (WT) DnaA and DnaC.

ΔpriC mutant were comparable to those of *priC*⁺. Similar results were observed in cells grown in M9 minimal medium at 30 °C. However, when grown in M9 minimal medium at 37 °C, ΔpriC mutant cells exhibited slightly reduced ori/mass values. Also, it is noteworthy that in all conditions we tested, the fraction of cells with non-$2^n$ *oriC* copies was slightly higher in ΔpriC cells compared to *priC*⁺. The asynchronous initiations would be caused by partial inhibition of initiation of multiple origins in ΔpriC cells, as representatively seen for the case of ΔpriC cells growing in M9 minimal medium at 37 °C where four-*oriC* cells reduced and three-*oriC* cells increased, compared to *priC*⁺ cells. A similar case might be shown for ΔpriC cells growing in LB medium at 25 °C in that eight-*oriC* cells reduced and asynchronous initiations (A.I. %) as well as four-*oriC* cells increased. As such, inhibition of replication initiation would occur intrinsically at low frequency in the WT *dnaA/dnaC* background. PriC function would be effective to suppress the inhibition, thereby ensuring replication initiation of multiple origins.

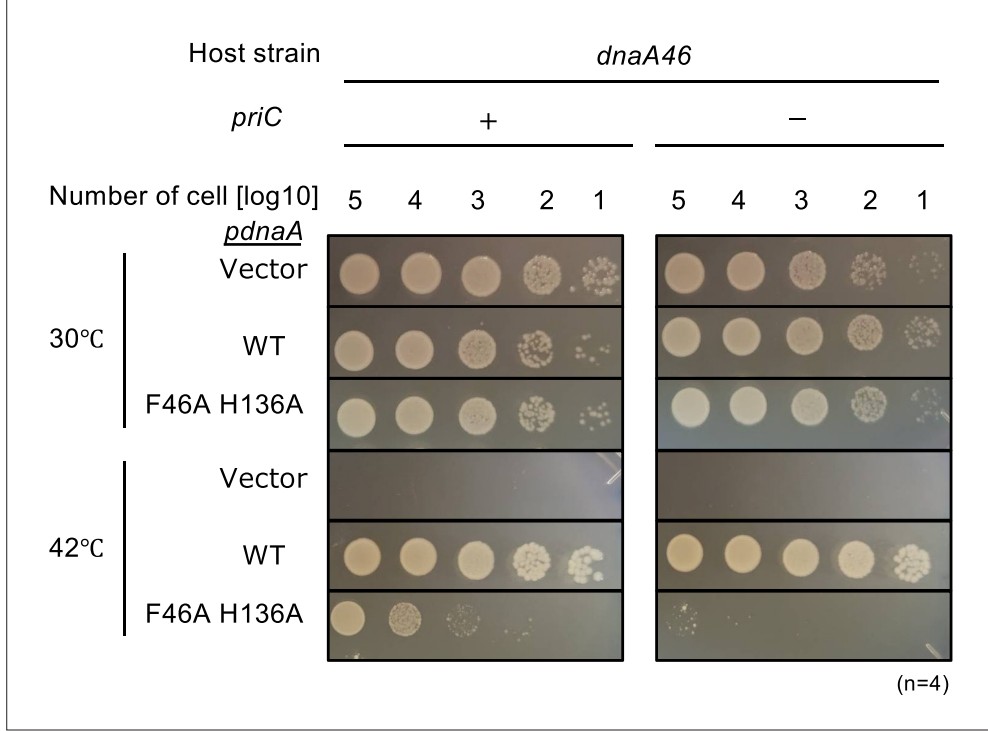

**Figure 3.** Role of PriC in growth of DnaA mutant defective in interactions with DnaB. Cell growth of MIT125 (MG1655 *dnaA46 tnaA*::Tn*10*) and KRC004 (MG1655 *dnaA46 tnaA*::Tn*10* Δ*priC::frt-kan*) cells bearing pING1 plasmid vector (-) or its derivatives pKA234 expressing wild-type (WT) DnaA (WT) or pFH expressing DnaA F46A H136A (F46A H136A). Cells were grown at 30 °C overnight and 10-fold serial dilutions of the cultures (~10⁹ cells/mL) were incubated on LB agar plates containing ampicillin for 16 hr at 30 °C and for 14 hr at 42 °C, respectively. Four independent experiments were performed. +, WT *priC*; -, Δ*priC::frt-kan*.

The online version of this article includes the following source data and figure supplement(s) for figure 3:

**Figure supplement 1.** Immunoblot analysis for plasmid-encoded DnaA proteins in *dnaA46* cells.

**Figure supplement 1—source data 1.** Original western blot corresponding to *Figure 3—figure supplement 1A*.

**Figure supplement 1—source data 2.** Original western blot corresponding to *Figure 3—figure supplement 1A*.

## PriC rescues the cell growth of DnaA mutants defective in DnaB interactions

DnaA contains two regions for interactions with DnaB during DnaB loading at *oriC* ssDNA: one is a high-affinity region in domain I containing Phe46, and the other is a low-affinity region in domain III containing His136. To analyze the mechanism of PriC-promoted DnaB loading at *oriC*, we introduced a series of pING1 vector-based plasmids encoding WT DnaA (pKA234) or the DnaA F46A H136A double mutant (pFH) into *dnaA46* cells with or without *priC*. Growth of *dnaA46* cells containing pING1 was inhibited at 42°C, but the leaky expression of *dnaA* from pKA234 enabled the growth of these cells, as previously reported (**Sakiyama et al., 2018**; **Figure 3**). Introduction of pFH moderately supported the cell growth of *dnaA46* cells at 42°C. By contrast, *dnaA46 ΔpriC* cells carrying pFH exhibited little growth at 42°C (**Figure 3**).

Western blotting analysis supported that the expression levels of WT DnaA and DnaA F46A H136A proteins in *dnaA46* cells bearing pKA234 or pFH were comparable, regardless of the presence or absence of *priC⁺* (**Figure 3—figure supplement 1**). As DnaA46 is unstable and thermolabile (**Shimuta et al., 2004**), these cells were grown at 30 °C, and were incubated at 42 °C for 90 min before Western blotting analysis. The DnaA level in *dnaA46* cells bearing the vector pING1 plasmid was minimum.

These results suggest that PriC is crucial for the growth of cells defective in specific DnaA-DnaB interactions, possibly because it can bypass the DnaA-DnaB interaction step required for DnaB loading onto *oriC* ssDNA.

## PriC rescues DNA replication inhibition by DiaA overexpression in vivo

DiaA, a homotetrameric protein, binds to multiple DnaA molecules and stimulates DnaA multimer assembly at *oriC* (**Keyamura et al., 2007**). The Phe46 residue in DnaA domain I is part of the DiaA-binding site and DnaB binding site. Oversupply of DiaA inhibits timely initiation of replication in cells growing in tryptone medium at 30 °C, probably because of binding competition between DiaA and DnaB for DnaA domain I (**Ishida et al., 2004**; **Keyamura et al., 2009**). If PriC can bypass the DnaA-DnaB interaction requirement for DnaB loading onto *oriC*, PriC should suppress inhibition of initiation by DiaA oversupply. To test this possibility, we introduced pBR322 or its derivative encoding *diaA* (pNA135) into cells with or without *priC* (**Figure 4**). *priC⁺* carrying pBR322 or pNA135 similarly grew on LB agar plates containing ampicillin between 25°C and 42°C (**Figure 4A**, **Figure 4—figure supplement 1**). However, cell growth of *ΔpriC* cells carrying pNA135, but not pBR322, was moderately inhibited at 25 °C (**Figure 4A**), supporting our conjecture that PriC can bypass the requirement for the DnaA-DnaB interaction.

Rep helicase stimulates PriC-mediated rescue of arrested replication forks containing nascent strands (**Heller and Marians, 2005b**; **Heller and Marians, 2005a**; **Sandler, 2000**). To test the contribution of Rep helicase to PriC-promoted rescue of blocked replication initiation, we introduced pBR322 or pNA135 into *Δrep* cells. Unlike *ΔpriC* cells, *Δrep* cells carrying pNA135 grew similarly to *Δrep* cells carrying pBR322 (**Figure 4A**), suggesting no involvement of Rep in the PriC-promoted rescue of blocked replication initiation. This is consistent with the features of unwound *oriC*, which has a fork structure similar to that of the replication fork but without nascent strands, and the specific role of Rep in rescuing abandoned replication forks with nascent strands in the PriC-dependent pathway. Rep is specifically required for the unwinding of the nascent leading strand of abandoned replication forks, which is needed to expand the ssDNA region required for entry of PriC (**Heller and Marians, 2005b**; **Heller and Marians, 2005a**).

We further analyzed replication initiation in *ΔpriC* cells carrying pNA135 at 30 °C by flow cytometry (**Figure 4B**). Under these conditions, *priC⁺* bearing pBR322 mainly had eight or four *oriC* copies per cell with eight-*oriC* cells predominating (**Figure 4B**). The number of *oriC* copies per cell was similar to that in *priC⁺*-bearing pNA135, with only slight increase in asynchronous initiation, as shown by the slight increase in the number of cells with non-$2^n$ *oriC* copies, which is basically consistent with our previous data (**Ishida et al., 2004**). However, in *ΔpriC* cells bearing pNA135, severe inhibition of initiation occurred, as shown by the substantial decrease in the number of eight-*oriC* cells and the large increase in the number of cells with only one to three *oriC* copies (**Figure 4B**). This result supports the proposed function of PriC at *oriC* mentioned above.

To test whether PriC rescues replication initiation inhibited by deletion of the *diaA* gene, we conducted similar flow cytometry analysis of *ΔpriC*, *ΔdiaA*, and *ΔpriC ΔdiaA* cells (**Figure 4C**). *ΔdiaA*



**Figure 4.** PriC stimulation of DNA replication initiation inhibited by DiaA oversupply. (**A**) Cell growth of MG1655, KRC002 (MG1655 Δ*priC::frt-kan*), and FF001 (MG1655 Δ*rep::frt-kan*) cells bearing pBR322 or pNA135 (pBR322 derivatives carrying *diaA*). Overnight cultures (~$10^9$ cells/mL) of MG1655 and KRC002 cells bearing pBR322 or pNA135 were 10-fold serially diluted and incubated on LB agar plates containing ampicillin for 40 hr at 25°C. The results of three independent experiments were consistent, and one which is shown. (**B**) Flow cytometry analyses of MG1655 and KRC002 (MG1655 Δ*priC::frt-kan*) cells bearing pBR322 or pNA135. Cells were exponentially grown at 30°C in LB medium with ampicillin, followed by further incubation with rifampicin and cephalexin to allow run-out of chromosomal DNA replication. DNA contents were quantified by flow cytometry, and cell size (mass) at the time of drug addition was measured by a Coulter counter. Mean mass, ori/mass ratio, doubling time (Td), and asynchrony index (A.I.: the percentage of cell numbers with non-$2^n$ copies of *oriC* per the cell numbers with $2^n$ copies of *oriC*) of each strain are indicated at the top right of each panel with standard deviations. Three to four independent experiments were performed. +, WT *priC*; -, Δ*priC::frt-kan*. (**C**) Flow cytometry analyses of MG1655, KRC003 (MG1655 Δ*priC::frt*), SA103 (MG1655 Δ*diaA::frt-kan*), and KRC006 (MG1655 Δ*priC::frt*, Δ*diaA::frt-kan*) cells. Cells were grown and analyzed

*Figure 4 continued on next page*

*Figure 4 continued*

as described above. Three independent experiments were performed. For '*priC*' column, +, WT *priC*; -, Δ*priC::frt*. For '*diaA*' column, +, WT *diaA*; -, Δ*diaA::frt-kan*.

The online version of this article includes the following figure supplement(s) for figure 4:

**Figure supplement 1.** Cell growth ability of cells carrying pBR322 or pNA135 with WT *priC* or Δ*priC::frt-kan*.

cells showed severe asynchrony of replication initiation, as shown by the increase in the number of five-to-seven-*oriC* cells, which supports the stimulatory role of DiaA in DnaA-*oriC* complex formation (*Figure 4C*). However, Δ*priC* Δ*diaA* cells did not show more asynchronization of replication initiation than Δ*diaA* cells, as shown by the similar asynchrony index (A.I.). These results suggest that PriC does not support or bypass DnaA-*oriC* complex formation.

## PriC stimulates replication initiation in cells with *oriC* sequence truncations

Right DnaA-subcomplex plays a stimulatory role in DnaB helicase loading (*Sakiyama et al., 2022*; *Stepankiw et al., 2009*). The loading of two DnaB hexamers at *oriC* ssDNA regions requires multiple steps that depend on the distinct functions of two DnaA subcomplexes (*Figure 1*): the Left-DnaA subcomplex stably unwinds M-R regions of the DUE and the Right-DnaA subcomplex expands the unwound region to the AT-L region, supporting efficient DnaB helicase loading (*Sakiyama et al., 2017*; *Sakiyama et al., 2022*; *Yoshida et al., 2023*). The expanded ssDNA regions in this open *oriC* complex efficiently promote the loading of one DnaB hexamer onto the lower (A-rich) strand M-R region and a second DnaB hexamer onto the upper (T-rich) strand of the DUE (*Fang et al., 1999*; *Sakiyama et al., 2022*). In the absence of the Right-DnaA subcomplex, the AT region of the AT-L region assists in DnaB loading (*Sakiyama et al., 2022*).

To determine whether PriC rescues defects in DnaB loading when the *oriC* sequence is altered to prevent Right-DnaA subcomplex formation, we first assessed the cell growth of cells lacking the Middle- and Right-DOR of *oriC* (Left-*oriC*) (*Figure 5A*). The Left-*oriC* mutant formed colonies at 37 °C but showed severe growth defects at 25°C and 30°C (*Figure 5B*). At 37 °C, thermal energy might be sufficient to expand DUE unwinding into the AT region and allow DnaB loading in the absence of the Right-DnaA subcomplex (*Sakiyama et al., 2022*), but not at 25 °C or 30 °C. By contrast, the cell growth of the Left-*oriC* Δ*priC* double mutant was markedly compromised at 37 °C and moderately reduced at 25°C and 30°C (*Figure 5B*). These findings support the idea that PriC functions to rescue defective processes in DnaB loading caused by deletions of the *oriC* sequence that prevent DnaA subcomplex formation.

Next, similar experiments were performed using a mutant in which the R4 DnaA box within Right-DOR was substituted with a sequence (R4*Tma*) defective in binding of *E. coli* DnaA (*Figure 5A and C*; *Noguchi et al., 2015*; *Sakiyama et al., 2022*). We reasoned that formation of the Right-DnaA subcomplex would be inhibited by the introduction of R4*Tma* and PriC would be required for robust initiation in this context. As the Left-*oriC* strain, the *priC*+ R4 *Tma* strain showed normal cell growth. By contrast, cell growth of the R4*Tma* Δ*priC* double mutant was impaired moderately at 25 °C and slightly at 30 °C (*Figure 5C*), indicating that PriC can rescue the blocked replication initiation caused by the absence of the R4 box. The reduced cell growth of R4*Tma* Δ*priC* strain was alleviated at higher temperatures, similar to the phenotype of the Left-*oriC* mutant.

Furthermore, we assessed replication initiation in R4*Tma* cells at 30 °C using flow cytometry (*Figure 5D*). R4*Tma* cells grew slightly slower than WT *oriC* cells and showed clear inhibition of initiation; in contrast to the WT *oriC* strain, the R4*Tma* mutant cell population contained more four-*oriC* cells and fewer eight-*oriC* cells, and showed severe asynchronous initiation. Notably, these negative effects of the R4*Tma* mutation were amplified by deletion of *priC*, i.e., the number of one to three-*oriC* cells increased and the ori/mass ratio was further reduced.

We next analyzed the role of PriC in cells lacking the AT-L region of *oriC* (subATL *oriC*) (*Figure 6A*). L-DUE and the flanking AT region assist in DnaB helicase loading by stimulating *oriC* unwinding, which is essential in a strain defective in Right-DnaA subcomplex formation (*Sakiyama et al., 2022*). SubATL *oriC* cells and WT *oriC* cells grew similarly on LB agar plates between 25°C and 37°C, regardless of the presence of Δ*priC* (*Figure 6B*). However, flow cytometry analysis revealed that deletion of *priC*

**A**

**B** LB

| Genome *oriC* | WT | | | | | | Left | | | | | |
|---|---|---|---|---|---|---|---|---|---|---|---|---|
| Number of cell [log10] | 6 | 5 | 4 | 3 | 2 | 1 | 6 | 5 | 4 | 3 | 2 | 1 |

(n=3)

**C** LB

| Genome *oriC* | WT | | | | | | R4*Tma* | | | | | |
|---|---|---|---|---|---|---|---|---|---|---|---|---|
| Number of cell [log10] | 6 | 5 | 4 | 3 | 2 | 1 | 6 | 5 | 4 | 3 | 2 | 1 |

(n=4)

**D** LB, 30℃

Genome *oriC*   WT   R4*Tma*

| | WT + | WT − | R4*Tma* + | R4*Tma* − | |
|---|---|---|---|---|---|
| | [1] | 0.97±0.10 | 1.10±0.13 | 1.36±0.11 | :mass |
| | [1] | 1.00±0.10 | 0.51±0.07 | 0.34±0.03 | :ori/mass |
| | 38±4 | 40±5 | 47±3 | 53±8 | :Td (min) |

Cell number — Chromosome number

(n=4)

**Figure 5.** Stimulation of replication initiation by PriC in Right-DnaA subcomplex-defective cells. (**A**) Schematic structure of Left-*oriC* and R4*Tma oriC*. AT-rich region, Duplex-Unwinding Element (DUE), and DnaA-Oligomerization Region (DOR) are indicated by a gray dotted box, a gray box, and an open box, respectively. Filled and open arrowheads in the DOR show DnaA boxes with the full consensus sequence or mismatches. In addition, a striped box indicates the IBR. The regions of each mutant *oriC* are indicated below the structure as open boxes. The position of the *Tma*DnaA box substitution is

*Figure 5 continued on next page*

*Figure 5 continued*

indicated by a yellow arrowhead. (**B**) Cell growth of WT *oriC* and Left-*oriC* cells with or without *priC*. Overnight cultures (~10⁹ cells/mL) of NY20-frt (WT *oriC*; WT) and NY20L-frt (Left-*oriC*; Left) cells with (+) or without (-) *priC* were 10-fold serially diluted and incubated on LB agar plates for 24 hr at 25 °C, for 16 hr at 30 °C, and for 14 hr at 37 °C. +, WT *priC*; -, Δ*priC::frt-kan*. Three independent experiments were performed. (**C**) Cell growth of WT *oriC* and *oriC* R4-box mutant cells with or without *priC*. NY20-frt (WT *oriC*) and NY24-frt (*oriC* with R4*Tma* substitution, R4*Tma*) cells with (+) or without (-) *priC* were grown as described in panel A. +, WT *priC*; -, Δ*priC::frt-kan*. Four independent experiments were performed. (**D**) Flow cytometry analyses of WT and R4*Tma* cells with or without *priC*. Cells were exponentially grown at 30℃ in LB medium with ampicillin and analyzed as described in *Figure 4*. Representative histograms from four independent experiments are shown. Mean mass, ori/mass, and Td of each strain are indicated at the top right of each panel with standard deviations.

specifically inhibited initiation in subATL *oriC* cells (*Figure 6C*); the number of eight-*oriC* cells was lower while the number of four-to-seven *oriC* cells was higher in subATL *oriC* Δ*priC* cells than in *priC⁺* *oriC* cells. Taken together, these results are consistent with the idea that PriC restores initiation and DnaB loading at such truncated *oriC* sequences.

## PriC loads DnaB at *oriC* unwound by the DnaA complex in an in vitro reconstituted system

To analyze the mechanism of PriC rescue of blocked initiation at *oriC*, we assessed PriC activity in an in vitro reconstituted system for DnaB loading and DnaB-promoted DNA unwinding using the supercoiled circular form (form I) of the *oriC* plasmid pBS*oriC* and purified proteins, namely DnaA, N-terminally histidine-tagged DnaB (His-DnaB), DnaC, IHF, SSB, and gyrase. In this system, DnaA and IHF unwind the DUE and stimulate His-DnaB loading to the ssDUE region with the aid of DnaC. The loaded His-DnaB expands the ssDNA regions in concert with the ssDNA-binding activities of SSB. Concomitantly, the positive supercoiling generated through DNA unwinding by His-DnaB is resolved by DNA gyrase, resulting in the formation of plasmid DNA topoisomers (form I*). Form I* can be separated from form I by agarose gel electrophoresis (*Baker et al., 1986*; *Sakiyama et al., 2022*).

First, we assessed the effect of PriC on WT DnaA. WT DnaA stimulated form I* formation in the absence of PriC (*Figure 7A*, lanes 1 and 5) and increasing concentrations of PriC moderately inhibited WT DnaA-mediated form I* formation (*Figure 7A*, lanes 5–8), probably because of competition between DnaA and PriC for binding to DnaB. Next, we focused on the DnaA F46A H136A double mutant. The DnaA F46A and DnaA H136A mutant proteins alone are reported not to support form I* formation (*Keyamura et al., 2009*; *Sakiyama et al., 2018*). Consistently, the DnaA F46A H136A double mutant protein was inactive in form I* formation (*Figure 7A*, lane 9). However, when PriC was added to the assay, DnaA F46A H136A promoted form I* formation to a level comparable to that of WT DnaA in the presence of PriC (*Figure 7A*, lanes 5–12 and 7B), indicating that PriC rescues defective DnaB loading by DnaA F46A H136A-*oriC* complexes. These findings are consistent with the idea that PriC can bypass the strict reliance on DnaA-DnaB interactions for DnaB loading at *oriC*.

To corroborate this idea, we also determined whether PriC rescues blocked initiation resulting from unstable DUE unwinding. *oriC* complexes containing DnaA V211A cannot bind to the ssDUE and consequently fails in stable DUE unwinding (*Ozaki et al., 2008*). The DnaA V211A mutant was virtually inactive for form I * formation irrespective of the presence or absence of PriC (*Figure 7A*, lanes 13–16, and 7B), indicating that PriC cannot efficiently rescue a defect of DUE unwinding.

We further evaluated the DnaB loading activity of PriC in the presence of DiaA using a similar in vitro reconstituted system. Previously, DiaA was shown to stimulate *oriC* DUE unwinding, but to inhibit form I* formation, because DiaA-DnaA binding competitively inhibits DnaB-DnaA binding; DiaA and DnaB share the same binding site in DnaA domain I (*Figure 7C* lanes 1, 3, 5, and 7 and 7D) (*Keyamura et al., 2009*). PriC moderately stimulated form I* formation specifically in the presence of both DnaA and DiaA (*Figure 7C* lanes 7 and 8, and 7D). These results further support the idea that PriC can bypass the specific requirement for the DnaA-DnaB interaction in DnaB loading at *oriC*. DiaA bound to *oriC*-DnaA complexes might be a physical obstacle reducing efficiency of DnaB loading by PriC.

In addition, we analyzed the DUE unwinding activities of DnaA mutants using the in vitro reconstituted system. As previously reported, DnaA V211A is largely inactive in DUE unwinding, but DnaA F46A H136A unwinds the DUE at a level comparable to that of WT DnaA (*Figure 7—figure supplement 1A and B*). PriC did not stimulate DUE unwinding irrespective of the presence of WT DnaA (*Figure 7—figure supplement 1C and D*). These results demonstrate the functional specificity of PriC

**A**

**B** LB

**C** LB, 30℃

**Figure 6.** Stimulation of replication initiation by PriC in cells with a deletion in the ssDNA-expanding region of *oriC*. (**A**) Schematic structure of subATL *oriC* as shown in *Figure 5*. The *oriC* region, including the subATL *oriC* is indicated below the structure as a white box. (**B**) Cell growth of wild-type (WT) *oriC* and *oriC* subATL mutant cells with or without WT *priC*. NY20-frt (WT *oriC*) and NY20ATL-frt (*oriC* lacking AT and L sequences, subATL) with (+) or without (-) WT *priC* were grown as described in *Figure 5*. +, WT *priC*; -, Δ*priC::frt-kan*. Three independent experiments were performed. (**C**) Flow cytometry analyses of WT *oriC* and *oriC* subATL mutant cells with or without WT *priC*. Cells were exponentially grown at 30℃ in LB medium and analyzed as described in *Figure 4*. Representative histograms from three independent experiments are shown. Mean mass, ori/mass, and Td of each strain are indicated at the top right of each panel with standard deviations.

in rescuing failed DnaB loading at *oriC* ssDNA and are consistent with the PriC mechanism mentioned above.

## Role for PriC in cSDR

Finally, to analyze the role of PriC in other types of replication initiation, we examine whether PriC contributes to the initiation of cSDR, which does not require *dnaA* or *oriC* and is activated in cells lacking *rnhA* or *recG*. Although the PriA-PriB-DnaT primosome complex has previously been reported to be involved in the initiation of cSDR (*Heller and Marians, 2006*), it is unclear whether PriC has any role in this type of initiation.

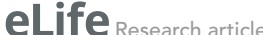

**Figure 7.** DnaB loading onto the ssDUE by PriC. (**A and B**) In vitro reconstituted system for DnaB loading. Form I of *oriC* plasmid (pBS*oriC*; 1.6 nM) was incubated for 15 min at 30℃ with the indicated amount of PriC in the presence of ATP-DnaA or its mutant derivatives, DnaB, DnaC, IHF, gyrase, and Single-Stranded DNA Binding protein (SSB). The resulting plasmids were purified and analyzed using agarose gel electrophoresis. A representative image of three independent experiments is shown in the black/white-inverted mode (**A**). In panel A, the size markers of linear dsDNA (M) are also

*Figure 7 continued on next page*

*Figure 7 continued*

shown. Band intensities of each lane in the gel image were analyzed by densitometric scanning. The percentages of form I* *oriC* plasmid per input DNA are shown as 'Form I* (%)'. Mean and standard deviations are shown (**B**). Abbreviations in panel B: -, no addition of DnaA; WT, wild-type DnaA; FH, DnaA F46A H136A; VA, DnaA V211A. (**C and D**) In vitro reconstituted system containing DiaA. Form I of pBS*oriC* plasmid was used as described above except for the addition of the indicated amounts of DiaA. A representative image of three independent experiments is shown in black/white-inverted mode (**C**). In panel C, the size markers of linear dsDNA (M) are also shown. Band intensities of each lane in the gel image were analyzed by densitometric scanning. The percentages of form I**oriC* plasmid together with the mean and standard deviations are shown as described above (**D**).

The online version of this article includes the following source data and figure supplement(s) for figure 7:

**Source data 1.** Original gels corresponding to *Figure 7*, panels A (Upper gel) and C (Lower gel).

**Source data 2.** Original gel image corresponding to *Figure 7A*.

**Source data 3.** Original gel image corresponding to *Figure 7B*.

**Figure supplement 1.** Duplex-Unwinding Element (DUE) unwinding activities of DnaA mutant derivatives in the presence and absence of PriC.

**Figure supplement 1—source data 1.** Original gels corresponding to *Figure 7—figure supplement 1A and C*.

**Figure supplement 1—source data 2.** Original gel image corresponding to *Figure 7—figure supplement 1A*.

**Figure supplement 1—source data 3.** Original gel image corresponding to *Figure 7—figure supplement 1C*.

Unlike *dnaA46* cells, *dnaA46 rnhA::cat* double mutant cells grew even at 40 °C (*Figure 8A*), indicating that these cells were engaged in cSDR, as previously reported (*Hinds and Sandler, 2004*). The cell growth of Δ*priC dnaA46 rnhA::cat* triple mutant was severely inhibited at 40 °C (*Figure 8A*), suggesting that PriC contributes to the growth of *dnaA46 rnhA::cat* mutant cells.

To determine whether PriC contributes to the initiation of *dnaA46-oriC* replication or cSDR, we calculated the relative copy numbers of *oriC* (84.6 min) and *ter* (32.4 min: corresponding to *insQ*) using real-time quantitative PCR (qPCR). The *ypaB* (50.5 min) located at the middle position between *oriC* and *ter* was used as a reference. To calculate the copy number of the chromosomal *ter* region, the preferential initiation site of cSDR (*Brochu et al., 2018*; *Maduike et al., 2014*), *dnaA46 rnhA::cat* cells growing exponentially at 30 °C were shifted to 40 °C for 90 min and the copy number before and after incubation at 40 °C was calculated. Consistent with the cell growth data (*Figure 8A*), the *ter* copy number ratio of the *dnaA46 rnhA::cat* double mutant increased after 40 °C incubation, whereas the copy number ratio of the *dnaA46* mutant expressing WT *rnhA* remained the same (*Figure 8B*). The *ter* copy number ratio of the *dnaA46 rnhA::cat* Δ*priC* triple mutant strain was also increased, indicating that cSDR could occur even in the absence of PriC.

To monitor initiation of DnaA-*oriC* replication, we determined the *oriC* copy number ratio (*Figure 8B*). At the permissive temperature, despite having similar growth rates, the *oriC* copy number ratio of the *dnaA46* Δ*priC* double mutant cells was lower than that of the *dnaA46* single mutant, confirming the importance of PriC for replication initiation in *dnaA46* cells, as shown in *Figure 2*. In the *priC+dnaA46 rnhA::cat* mutant cells, the *oriC* copy number ratio was reduced, which is explained by activation of cSDR and increase in the relative copy number of the reference *ypaB* locus (50.5 min chromosomal map position). Together, these results suggest the idea that PriC stimulates replication fork progression during cSDR.

To consolidate these findings, we performed the chromosome loci copy-number analysis using whole-genome sequencing. The *dnaA46* cells and its derivative cells were grown exponentially at 30 °C, followed by incubation at 40 °C for 90 min prior to DNA extraction. In growing WT cells, the highest copy numbers are observed near the *oriC*-proximal region and the lowest are near the *ter*-proximal region, resulting in a coverage bias from *oriC* to *ter* (*Brochu et al., 2018*; *Maduike et al., 2014*). However, in *dnaA46* cells incubated at 40 °C, the coverage bias was largely diminished due to the run-out replication (*Figure 8—figure supplement 1A*). Notably, *dnaA46 rnhA::cat* cells incubated at 40 °C exhibited discrete coverage peaks at the preferential initiation sites of cSDR near the *ter* (or *insQ*) region with a decreasing coverage bias to *oriC*. Similar profiles were observed in the Δ*priC* mutant background, supporting the idea that cSDR can occur even in the absence of PriC (*Figure 8—figure supplement 1A*). Moreover, in the *dnaA46 rnhA::cat* background, the absence of *priC* appeared to decrease the coverage level at genomic positions near the 1.3 MB and 4 MB map positions, which are slightly outside of *ter* and *oriC*, respectively (*Figure 8—figure supplement 1B*). This observation may reflect the role of *priC* in the replication fork progression.

**A**

LB (15 mg/ml tetracycline)

Host strain: *dnaA46*

[Figure 8A: Spot dilution assay plates at 30°C/18 hr and 40°C/36 hr with cell numbers 6 5 4 3 2 1, showing rnhA and priC combinations]

**B**

[Figure 8B: Bar graph showing Relative copy number (/ypaB) for oriC and ter]

**Figure 8.** PriC stimulation of constitutive stable DNA replication (cSDR)-dependent cell growth. (**A**) Cell growth abilities. Cells of MIT125 (MG1655 *dnaA46*), MIT125c (MG1655 *dnaA46 rnhA::cat*), KRC004 (MG1655 *dnaA46 ΔpriC::frt-kan*), and KRC004c (MG1655 *dnaA46 ΔpriC::frt-kan rnhA::cat*) were grown at 30°C overnight and 10-fold-serial dilutions of the overnight cultures (~10⁹ cells/mL) were incubated for 16 h at 30°C and for 36 hr at 40°C on LB agar plates. Three independent experiments were performed. +, WT; -, deletion. (**B**) *oriC* and *ter* copy numbers. MIT125, MIT125c, KRC004, and KRC004c cells growing exponentially at 30°C were further incubated for 90 min at 40°C in LB medium. Samples were withdrawn before and after the 40°C incubation. The genome DNA of each sample was extracted by boiling the cells for 5 min at 95°C. The relative copy numbers of *oriC* (84.6 min) and *ter* (32.4 min) to that of *ypaB* (50.5 min) were quantified using real-time qPCR. The data and averages of three independent experiments are shown with standard deviations.

The online version of this article includes the following figure supplement(s) for figure 8:

**Figure supplement 1.** Chromosome loci copy-number analysis of *dnaA46* mutant cells and its derivative cells.

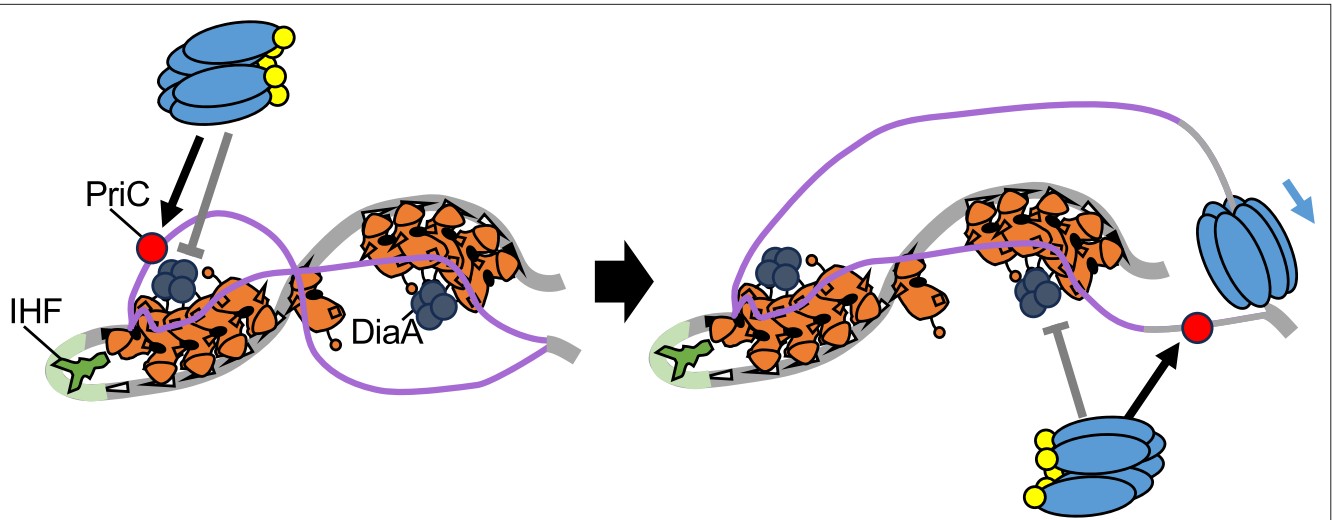

**Figure 9.** Model for PriC-promoted replication initiation. PriC-promoted rescue of blocked DnaA-*oriC*-dependent replication initiation. *oriC*, DnaA, IHF, DnaB, and DnaC are illustrated as depicted in *Figure 1C*. When tight DnaA binding of DiaA (dark gray circle) (or DnaA F46A H136A double mutations) inhibits DnaB recruitment, PriC (Brilliant red circle) binds to a stably unwound DUE strand, recruiting a DnaB-DnaC complex to lower strand of DUE instead of Left-DnaA subcomplex. When a sufficient ssDNA region is exposed, PriC then recruits the DnaB-DnaC complex to the opposite strand.

## Discussion

In growing cells, replication initiation can be blocked under challenging conditions. However, the mechanisms that rescue blocked initiations are largely unknown. To elucidate such mechanisms, we investigated PriC-promoted replication initiation under various challenging conditions. We showed that PriC was necessary for the optimal growth of *dnaC2* cells at the semi-restrictive temperature and for replication initiation even at the permissive temperature (*Figure 2*), suggesting that PriC contributes to replication initiation rescue in *dnaC2* cells by facilitating DnaB helicase loading at *oriC*. Furthermore, we observed that PriC was necessary for the optimal growth of cells in which the DnaA-DnaB interaction was inhibited by DiaA overexpression or DnaA F46A H136A double muta-tions (*Figures 3 and 4*). In addition, PriC was stimulatory for the cell growth and replication initiation bearing *oriC* mutations that impair efficient DnaB loading (*Figures 5 and 6*). These results, together with results of form I* formation assays (*Figure 7*), suggest that PriC bypasses the need for the DnaA-DnaB interaction by binding to unwound *oriC* DNA and recruiting DnaB to facilitate its loading onto ssDNA (*Figure 9*). This mechanism is similar to that proposed for the rescue of abandoned replication forks by PriC (*Figure 1E*).

Also, PriC contributes to replication initiation rescue in *dnaA46* cells at 37 °C. It is presumable that DnaA46 protein becomes partially denatured at the sub-permissive temperature of 37 °C (*Hwang and Kaguni, 1988*; *Carr and Kaguni, 1996*). This partial denaturation should impair both origin unwinding and helicase loading, though not to the extent that cell viability is lost. The *priC* deletion should further exacerbate helicase loading defects by inhibiting the bypass mechanism (*Figure 9*), resulting in the lethality of *dnaA46* cells at this temperature.

There might remain an alternative possibility that PriC could stabilize the DnaA-DnaB interaction for rescuing the impaired DnaB loading process. However, we believe that this possibility is not very likely. Given that interactions between DnaA and DnaB during DnaB loading to *oriC* are highly dynamic and involve regulated multiple steps, the stabilization of the DnaA-DnaB interaction by PriC, even if it occurs, carries a considerable risk of inhibiting the DnaB loading by constructing abortive complexes. In addition, DnaA-DiaA binding is very tight and stable (*Keyamura et al., 2007*; *Keyamura et al., 2009*). Even if WT DnaA and WT DnaB are present, PriC can rescue the initiation defects of *oriC* mutants. Based on these facts and the known characteristics of PriC, it is more reasonable to infer that PriC provides a bypass of DnaB loading at *oriC*, similar to the mechanism at the stalled replication fork (*Figures 1E and 9*).

The finding that PriC failed to stimulate form I* formation in the presence of DnaA V211A, which lacks DUE unwinding activity (*Figure 7*), provides mechanistic insight into PriC-dependent replication

initiation at *oriC*. We propose that PriC-dependent replication initiation possibly begins with the recognition of a stably unwound ssDNA region by PriC (*Figure 9*). Consistent with this perspective, PriC ssDNA-binding activity and ssDNA regions have been reported to be important for PriC-promoted replication fork rescue (*Heller and Marians, 2006*; *McMillan and Keck, 2024*). The failure of PriC to stimulate replication initiation in Δ*diaA* cells (*Figure 4C*) is also potentially explained by the reduced DUE unwinding caused by destabilization of the DnaA-*oriC* complex in Δ*diaA* cells. Based on this, we suggest that the stably unwound ssDNA region in *oriC* serves as a distinctive feature for PriC-promoting DnaB loading at *oriC* for replication initiation rescue, and distinguishes it from PriA-dependent replication restart, which requires the specific structure of the replication fork DNA consisting of sister dsDNA strands (*Heller and Marians, 2005a*; *Windgassen et al., 2018*). Unlike PriA, PriC can interact with the simple ssDNA to reload DnaB (*Figure 1E*).

In this study, we found that cells with DnaA F46A H136A double mutations grew well, but only when the cells expressed PriC. As for the DnaA F46A single mutant, our previous studies show that DnaA F46A has a limited residual activity in vivo (unlike in vitro), and allows slow growth of cells (*Abe et al., 2007*). The growth of cells expressing the DnaA H136A single mutant was severely inhibited even when the cells expressed PriC (*Sakiyama et al., 2018*). The His136 residue is located within the weak, secondary DnaB interaction region in DnaA, and is crucial for DnaB loading onto *oriC* ssDNA. Although domain I in DnaA H136A can stably tether DnaB-DnaC complexes to DnaA complexes on *oriC* (*Sakiyama et al., 2018*), the complexes fail to load DnaB onto *oriC* ssDNA even in the presence of PriC. It is possible that the interaction between PriC and DnaB is inhibited by stable DnaB binding to DnaA domain I. This idea is consistent with the in vivo feature of DnaA F46A single mutant. Conversely, the PriC-DnaB interaction may inhibit the interaction between DnaA domain I and DnaB, as suggested by the inhibitory effect of PriC on DnaB loading in vitro in the presence of WT DnaA (*Figure 7*). However, this inhibitory effect of PriC was not observed in vivo, suggesting that PriC-DnaB binding could involve still unknown factors and regulatory mechanisms.

PriC supported cSDR-dependent growth in *dnaA46 rnhA::cat* double mutant cells without stimulating the initiation of cSDR (*Figure 8*). In this mutant, cSDR predominantly initiates from the *ter* region, resulting in head-on collisions between the replisome and transcription complexes in *rrn* operons (*Maduike et al., 2014*). This suggests that in cSDR cells, PriC specifically rescues abandoned forks, including those generated during head-on conflicts between the replisome and transcription complexes.

*Bacillus subtilis* (*B. subtilis*), which expresses a PriA homolog but not a PriC homolog, employs a PriA-independent fork rescue pathway (*Bruand et al., 2001*). In this pathway, *B. subtilis* DnaC (*Bsu*DnaC) helicase is reloaded using the helicase loader DnaI together with *B. subtilis* DnaB (*Bsu*DnaB) and DnaD coloaders. The ssDNA-binding activity of *Bsu*DnaB has a crucial role in this PriA-independent pathway (*Bruand et al., 2005*). SsDNA-bound *Bsu*DnaB remodels the SSB-ssDNA complex with DnaD and recruits *Bsu*DnaC. These functions of *Bsu*DnaB are similar to the proposed functions of PriC in replication fork rescue in *E. coli*. In addition, *Bsu*DnaC helicase is recruited to *oriC* by concerted actions of *B. subtilis* DnaA and the DnaD-*Bsu*DnaB complex, which has an essential role in chromosome replication initiation (*Jameson and Wilkinson, 2017*), suggesting a similar role of *Bsu*DnaB to PriC in *E. coli* cells expressing DnaA F46A H136A. These similarities between PriC and *Bsu*DnaB in these two evolutionarily distant bacterial species suggest that the mechanism of PriC-promoted helicase loading is conserved among bacterial species despite the absence of sequence conservation of PriC and *Bsu*DnaB homologs.

Based on our results, we propose the abnormal competition between DnaB and DiaA for DnaA domain I could represent a form of intrinsic replication initiation stress in bacteria with conserved DiaA homologs. This type of stress could also occur in ε-proteobacterial species, such as *Helicobacter pylori*, because they express HobA, a DiaA-functional homolog (*Natrajan et al., 2007*; *Zawilak-Pawlik et al., 2011*). Similarly, YfdR, encoded by *E. coli* prophages, inhibits replication initiation by competing with DnaB for binding to DnaA domain I (*Noguchi and Katayama, 2016*), suggesting that such extraneously introduced inhibitors could trigger replication initiation stress.

Even in humans, ORC1, ORC2, and ORC5, the essential components of the eukaryotic replication initiation complex, are not essential in some cancer cell lines (*Shibata and Dutta, 2020*; *Shibata et al., 2016*), suggesting that mechanisms of replication initiation rescue may also operate beyond the bacterial kingdom. Therefore, further investigation of the initiation rescue processes and the factors

involved in diverse organisms from bacteria to human will be important for a full understanding of the common principles and diverse mechanisms that ensure robust initiation of chromosomal DNA replication.

## Materials and methods

### Plasmids, proteins, and strains

The plasmids used in this study are listed in key resource table. pKA234, pKW44-1, pNA135, pBSoriC, pET22b(+)-priC, and pTKM601 were described previously (*Aramaki et al., 2015*; *Ishida et al., 2004*; *Kawakami et al., 2005*; *Keyamura et al., 2009*; *Kubota et al., 1997*; *Ozaki et al., 2008*). For the construction of pFH, an alanine substitution was introduced into pH136A with specific mutagenic primer sets using the QuikChange site-directed mutagenesis protocol (Stratagene [Agilent], Agilent, La Jolla, CA, United States) as described previously (*Keyamura et al., 2007*; *Sakiyama et al., 2018*).

WT DnaA and its derivative proteins were overproduced in *E. coli* strain KA450 from pKA234, pKW44-1, or pFH and purified as described previously (*Noguchi et al., 2015*; *Ozaki et al., 2008*; *Sakiyama et al., 2017*).

PriC protein was prepared as previously reported with minor modifications (*Aramaki et al., 2013*). Briefly, PriC protein was overproduced in *E. coli* strain BL21-codonPlus-RIL by inducing its expression from pET22b(+)-priC using 0.2 mM isopropyl-β-D-1-thiogalactopyranoside (IPTG). The resulting cells were suspended in chilled buffer A (50 mM HEPES-NaOH [pH7.0], 10% sucrose, 1 mM EDTA, 2 mM dithiothreitol (DTT), and 1 mM PMSF) and disrupted by sonication. Proteins in the soluble fraction were precipitated with 0.24 g/mL ammonium sulfate, resuspended in a separation buffer (50 mM imidazole [pH 7.0], 20% glycerol, 2 mM DTT, 1 mM EDTA, and 40 mM ammonium sulfate) and loaded onto a 1 mL HiTrap SP HP column (Cytiva, Uppsala, Sweden). Bound proteins were eluted with a linear gradient from 0 to 1 M sodium chloride.

All *E. coli* strains used in this study are listed in key resource table. KYA018, MIT125, MIT162, SA103, NY20, and NY21 were described previously (*Kasho and Katayama, 2013*; *Noguchi et al., 2015*; *Noguchi and Katayama, 2016*). Strains bearing mutant *oriC* (NY20L and NY20ATL) were constructed using the λ Red site-directed recombination system as previously described (*Noguchi et al., 2015*; *Sakiyama et al., 2017*; *Sakiyama et al., 2022*). NY20-frt strain and *oriC* mutant strains (NY21-frt, NY20L-frt, and NY20ATL-frt) were constructed by eliminating *kan* from the NY20 strain and strains NY21, NY20L, and NY20ATL. To remove the kanamycin-resistant cassette (*kan*) from *oriC*, FLP recombinase encoded on pCP20 was used. Elimination of *kan* was verified by checking sensitivity to 50 µg/ml kanamycin in LB agar plates. ΔpriC::frt-kan from JW0456-KC and Δrep::frt-kan from JW5604-KC were introduced into MG1655 by P1 transduction, resulting in the construction of KRC002 and FF001, respectively. For construction of other ΔpriC::frt-kan strains (KRC004, KRC005, NY20-priC, NY21-priC, NY20L-priC, NY20ATL-priC), P1 phage lysates prepared from KRC002 were used for transduction of strains MIT125, KYA018, NY20-frt, SYM21-frt, NY20L-frt, and NY20ATL-frt. Transductants were screened on LB agar plates containing 50 µg/mL kanamycin. ΔdiaA::frt-kan or rnhA::cat derivatives (KRC006, MIT125c, and KRC004c) were constructed using a similar protocol except that P1 phage lysates from SA103 (ΔdiaA::frt-kan) or MIT162 (rnhA::cat) were used. KRC003 was generated by removal of *kan* from KRC002.

### Buffers

Buffer P contained 60 mM HEPES–KOH (pH 7.6), 0.1 mM zinc acetate, 8 mM magnesium acetate, 30% [v/v] glycerol, and 0.32 mg/mL bovine serum albumin (BSA). Buffer N contained 50 mM HEPES–KOH (pH 7.6), 2.5 mM magnesium acetate, 0.3 mM EDTA, 7 mM DTT, 0.007% [v/v] Triton X-100, and 20% [v/v] glycerol. Form I* buffer contained 20 mM Tris-HCl (pH 7.5), 125 mM potassium glutamate, 10 mM magnesium acetate, 8 mM DTT, and 0.5 mg/mL BSA.

### DUE unwinding assay

DUE unwinding assays were performed essentially as described with minor modifications (*Sakiyama et al., 2022*; *Yoshida et al., 2023*). Briefly, DnaA was incubated with ATP in buffer N on ice for 3 min to generate ATP-DnaA. The indicated amount of PriC and ATP-DnaA or its mutant derivatives was incubated for 9 min at 30°C in 10 µL buffer P containing 5 mM ATP, 125 mM potassium glutamate,

1.6 nM pBSoriC, and 42 nM IHF, followed by further incubation with 1.5 units of P1 nuclease (Wako) for 5 min. The reaction was stopped by the addition of 1% SDS and 25 mM EDTA, and DNA was purified by phenol-chloroform extraction and ethanol precipitation. One-third of each purified DNA was digested with AlwNI (NEB), which yielded 2.6 kb and 1.0 kb fragments after DUE unwinding of pBSoriC. The resultant DNA fragments were analyzed by 1% agarose gel electrophoresis with 1x Tris-acetate-EDTA buffer for 30 min at 100 V, followed by ethidium bromide staining. Gel images were taken using the GelDoc GO imaging system (Bio-Rad Laboratories, Hercules, CA), and products derived from unwound plasmids were quantified using ImageJ software.

## Form I* assay

This assay was performed as previously described with minor modifications (*Baker et al., 1986*; *Noguchi et al., 2015*; *Sakiyama et al., 2022*). The indicated amount of ATP-DnaA was incubated for 15 min at 30°C in 12.5 µL of Form I* buffer containing 3 mM ATP, 1.6 nM pBSoriC, 42 nM IHF, 400 nM His-DnaB, 400 nM DnaC, 76 nM GyrA, 100 nM His-GyrB, and 760 nM SSB. The reaction was stopped by the addition of 0.5% SDS, and DNA was purified by phenol-chloroform extraction. Samples were analyzed by 0.65% agarose gel electrophoresis with 0.5x Tris-borate-EDTA buffer for 15 hr at 23 V, followed by ethidium bromide staining.

## Flow cytometry analysis

Flow cytometry analysis was performed essentially as described (*Kasho et al., 2014*). Briefly, cells were grown at 30°C in LB medium until the absorbance of the culture ($A_{600}$) reached 0.1. Portions of the cultures were diluted a thousand-fold into 5 ml LB medium and incubated at 30°C until the absorbance of the culture ($A_{600}$) reached 0.1. The remaining portions were further incubated to determine the doubling time (Td) by measuring $A_{600}$ every 20 min. At $A_{600}$ 0.1, aliquots of the cultures were fixed in 70% ethanol to analyze cell mass using the Multisizer 3 Coulter counter (Beckman Coulter, Brea, CA). The remaining cultures were further incubated for 4 hr with 0.3 mg/mL rifampicin and 0.01 mg/mL cephalexin to allow run-out replication of chromosomal DNA. The resultant cells were fixed in 70% ethanol. After DNA staining with SYTOX Green (Thermo Fisher Scientific, Waltham, MA), cellular DNA contents were analyzed on a FACS Calibur flow cytometer (BD Bioscience, Franklin Lakes, NJ). The number of the origins/cell (ori/cell) was determined from the histograms of flow cytometry analysis. The number of ori/cell was divided by the mean cell mass determined by cell mass analysis, resulting in ori/mass ratios. The numbers of cells containing non-$2^n$ copy number of *oriC* were divided by the number of cells containing $2^n$ copy number of *oriC* to calculate the asynchrony index (A.I.). For the analysis of cells bearing plasmids, LB medium was supplemented with 100 µg/mL ampicillin.

## Replication initiation frequency test using synchronized cultures

MIT125 (*dnaA46 tnaA*::Tn*10*) cells or its Δ*priC* derivative KRC004 were grown at 30°C, a permissive temperature, in LB medium until the absorbance of the culture ($A_{600}$) reached 0.04, followed by further incubation for 90 min at 42°C, a restrictive temperature. Aliquots of the cultures were fixed in 70% ethanol as 'Synchronization' samples. The remaining cultures were further incubated for 5 min to initiate DNA replication, followed by further incubation for 4 hr at 42°C with 0.3 mg/mL rifampicin and 0.01 mg/mL cephalexin to allow run-out replication of chromosomal DNA. The resultant cells were fixed in 70% ethanol as '5 min release' samples. After DNA staining with SYTOX Green (Life Technologies), cellular DNA contents were analyzed on a FACS Calibur flow cytometer (BD Bioscience). Ratios of the origins/cell (ori/cell) were determined from the histograms of flow cytometry analysis.

For synchronization of *dnaC2* mutant cells (KYA018 and KRC005), cells were grown in LB medium at 30°C until the $A_{600}$ reached 0.04, followed by further incubation for 80 min at 37°C, at the restrictive temperature. The resulting synchronized cultures were released for 5 min at 30°C and incubated in the presence of 0.3 mg/mL rifampicin and 0.01 mg/mL cephalexin at 30°C to allow run-out of chromosomal DNA replication.

## qPCR for analysis of cSDR

Cells of MIT125 (*dnaA46 tnaA*::Tn*10*) cells and MIT125c (*dnaA46 tnaA*::Tn*10 rnhA*::*cat*) or its Δ*priC* derivative KRC004 (*dnaA46 tnaA*::Tn*10* Δ*priC*::*frt-kan*) and KRC004c (*dnaA46 tnaA*::Tn*10 rnhA*::*cat* Δ*priC*::*frt-kan*) were grown at 30°C in LB medium until the absorbance of the culture ($A_{600}$) reached 0.04

and aliquots of the cultures were withdrawn. The remaining samples were further incubated for 90 min at 40°C and aliquots of the cultures were withdrawn. These samples were boiled for 5 min at 95°C and the genome DNA was extracted. The levels of *oriC* (84.6 min), *ter* (32.4 min), and *ypaB* (50.5 min) were quantified by real-time qPCR using TB Green Premix Ex TaqII (Tli RNaseH Plus) (TaKaRa, Shiga, Japan) and the following primers: ORI_1 and KWoriCRev for *oriC*, qoriK fw, and qoriK rev for *ter*, STM419, and STM420 for *ypaB*.

## Chromosome loci copy-number analysis by whole-genome sequencing

The analysis was performed based on a previous paper with modifications (*Brochu et al., 2018*). Cells of the *dnaA46* mutant derivatives MIT125 (*dnaA46 tnaA*::Tn*10*), MIT125c (*dnaA46 tnaA*::Tn*10 rnhA::cat*), KRC004 (*dnaA46 tnaA*::Tn*10 ΔpriC::frt-kan*) and KRC004c (*dnaA46 tnaA*::Tn*10 rnhA::cat ΔpriC::frt-kan*) were grown exponentially at 30°C in LB medium (50 mL). When the absorbance of the culture ($A_{600}$) reached 0.04, cells were incubated for 90 min at 40°C and collected by centrifugation. Cell pellets were dissolved by incubation at 37°C for 1 hr in TE buffer (0.6 mL) containing 0.5% SDS and 25 μg/ml proteinase K, and were mixed with 0.2 mL of buffer containing 4% cetyltrimethylammonium bromide, followed by further incubation for 10 min at 65°C, phenol-chloroform extraction and isopropanol precipitation. Precipitates were resuspended in Tris (pH7.5) buffer containing 0.1 mg/ml RNase A, followed by incubation for 1 hr at 37°C and DNA purification by FastGene Gel/PCR Extraction Kit (Nippon Genetics, Tokyo, Japan). The quality control of genomic DNA, library preparation, and whole-genome sequencing were performed by Rhelixa, Inc using NEBNext Ultra II DNA Library Prep Kit and Illumina NovaSeq X Plus. The number of reads (150 bp × two pair-ended) was at least 6.7 M (3.3 M pairs) per sample. The sequence data were analyzed using Galaxy (https://usegalaxy.org) and the coverage plot was generated essentially as described (*Ivanova et al., 2015*). Briefly, paired-end reads were mapped using the MG1655 reference genome (NC_000913) and the BWA-MEM algorithm, followed by calculation of the number of reads per genome coverage. DNA-sequence data are deposited in the NCBI Sequence Read Archive (Accession number: PRJNA1222470).

## Immunoblot analysis

The assay was performed as described previously with minor modifications (*Kawakami et al., 2005*; *Sakiyama et al., 2017*; *Yoshida et al., 2023*). Briefly, MIT125 (*dnaA46 tnaA*::Tn*10*) and KRC004 (*dnaA46 tnaA*::Tn*10 ΔpriC::frt-kan*) cells bearing pING1, pKA234, or pFH were grown at 30 °C in LB medium including 100 μg/ml ampicillin until the absorbance of the culture ($A_{600}$) reached 0.05, followed by incubation at 42 °C for 90 min. Cells were harvested by centrifugation and dissolved in SDS sample buffer to adjust the calculated $A_{600}$ of 0.5. Proteins were separated using SDS-12% polyacrylamide gel electrophoresis and were transferred to polyvinylidene difluoride membranes (Millipore) using a semi-dry blotting method (Bio-Rad). After blocking at 30 °C for 30 min in 3% gelatin-TBS (3% gelatin, 20 mM Tris–HCl, 500 mM NaCl, pH 7.5), the membrane was incubated overnight at 4 °C in 1% gelatin-TTBS (20 mM Tris–HCl, 500 mM NaCl, 0.05% Tween-20, pH 7.5) containing anti-DnaA rabbit antiserum (1:3000). After washing, the membrane was incubated at 30 °C for 1 hr in 1% gelatin-TTBS containing goat anti-rabbit IgG antibody conjugated to alkaline phosphatase (Bio-Rad), followed by development using AP Conjugate substrate Kit (Bio-Rad).

## Acknowledgements

We are grateful to Drs. Yusuke Akama and Yukari Sakiyama for initial exploratory study related to this work. Japan Society for the Promotion of Science (JSPS KAKENHI) [JP17H03656, JP20H03212, and JP23K27131].

# Additional information

## Funding

| Funder | Grant reference number | Author |
|---|---|---|
| Japan Society for the Promotion of Science | JP17H03656 | Tsutomu Katayama |
| Japan Society for the Promotion of Science | JP20H03212 | Tsutomu Katayama |
| Japan Society for the Promotion of Science | JP23K27131 | Tsutomu Katayama |

The funders had no role in study design, data collection and interpretation, or the decision to submit the work for publication.

## Author contributions

Ryusei Yoshida, Validation, Investigation, Writing – original draft, Writing – review and editing; Kazuma Korogi, Qinfei Wu, Validation, Investigation, Writing – review and editing; Shogo Ozaki, Tsutomu Katayama, Supervision, Funding acquisition, Validation, Investigation, Writing – original draft, Writing – review and editing

## Author ORCIDs

Ryusei Yoshida ![ORCID] https://orcid.org/0009-0005-3545-7793
Shogo Ozaki ![ORCID] https://orcid.org/0000-0002-4384-1366
Tsutomu Katayama ![ORCID] https://orcid.org/0000-0001-9994-1684

Reviewer #1 (Public review): https://doi.org/10.7554/eLife.103340.3.sa1
Reviewer #2 (Public review): https://doi.org/10.7554/eLife.103340.3.sa2
Reviewer #3 (Public review): https://doi.org/10.7554/eLife.103340.3.sa3
Author response https://doi.org/10.7554/eLife.103340.3.sa4

# Additional files

## Supplementary files
MDAR checklist

## Data availability

All data generated or analysed during this study are included in the manuscript and supporting files; source data files have been provided for Figure 7, Figure 3-figure supplement 1A, and Figure 7-figure supplement 1A and C.

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

# Appendix 1

**Appendix 1—key resources table**

| Reagent type (species) or resource | Designation | Source or reference | Identifiers | Additional information |
|---|---|---|---|---|
| Strain, strain background (*Escherichia coli*) | BW25113 Δ*priC*::*frt-kan* | Keio collection | JW0456-KC | |
| Strain, strain background (*Escherichia coli*) | BW25113 Δ*rep*::*frt-kan* | Keio collection | JW5604-KC | |
| Strain, strain background (*Escherichia coli*) | MG1655 *dnaC2 zjj18*::*cat* | *Kasho and Katayama, 2013* | KYA018 | |
| Strain, strain background (*Escherichia coli*) | MG1655 *dnaA46 tnaA*::Tn10 | *Noguchi and Katayama, 2016* | MIT125 | |
| Strain, strain background (*Escherichia coli*) | MG1655 *rnhA*::*cat* | *Noguchi and Katayama, 2016* | MIT162 | |
| Strain, strain background (*Escherichia coli*) | MG1655 Δ*diaA*::*frt-kan* | *Noguchi and Katayama, 2016* | SA103 | |
| Strain, strain background (*Escherichia coli*) | MG1655 Δ*priC*::*frt-kan* | This work | KRC002 | MG1655 x P1 JW0456-KC |
| Strain, strain background (*Escherichia coli*) | MG1655 Δ*priC*::*frt* | This work | KRC003 | KRC002 x pCP20 |
| Strain, strain background (*Escherichia coli*) | MIT125 Δ*priC*::*frt-kan* | This work | KRC004 | MIT125 x P1 JW0456-KC |
| Strain, strain background (*Escherichia coli*) | KYA018 Δ*priC*::*frt-kan* | This work | KRC005 | KYA018 x P1 JW0456-KC |
| Strain, strain background (*Escherichia coli*) | KRC003 Δ*diaA*::*frt-kan* | This work | KRC006 | KRC003 x P1 SA103 |
| Strain, strain background (*Escherichia coli*) | MG1655 Δ*rep*::*frt-kan* | This work | FF001 | MG1655 x P1 JW5604-KC |
| Strain, strain background (*Escherichia coli*) | MG1655 *asnA*::*frt-kan* | *Noguchi et al., 2015* | NY20 | |
| Strain, strain background (*Escherichia coli*) | MG1655 *asnA*::*frt-kan oriC*ΔR4 box::*Tma*DnaA box | *Noguchi et al., 2015* | NY21 | |
| Strain, strain background (*Escherichia coli*) | MG1655 *asnA*::*frt-kan oriC*ΔMiddle-Right DOR | This work | NY20L | |
| strain, strain background (*Escherichia coli*) | MG1655 *asnA*::*frt-kan oriC*ΔAT-L region | This work | NY20ATL | |

*Appendix 1 Continued on next page*

*Appendix 1 Continued*

| Reagent type (species) or resource | Designation | Source or reference | Identifiers | Additional information |
|---|---|---|---|---|
| Strain, strain background (*Escherichia coli*) | MG1655 *asnA::frt* | *Yoshida et al., 2023* | NY20-frt | |
| Strain, strain background (*Escherichia coli*) | MG1655 *asnA::frt oriC*ΔR4 box::*Tma*DnaA box | This work | NY21-frt | NY21 x pCP20 |
| Strain, strain background (*Escherichia coli*) | MG1655 *asnA::frt oriC*ΔMiddle-Right DOR | This work | NY20L-frt | NY20L x pCP20 |
| Strain, strain background (*Escherichia coli*) | MG1655 *asnA::frt oriC*ΔAT-L region | This work | NY20ATL-frt | NY20ATL x pCP20 |
| Strain, strain background (*Escherichia coli*) | NY20-*frt* Δ*priC::frt-kan* | This work | NY20-*priC* | NY20-frt x P1 KRC002 |
| Strain, strain background (*Escherichia coli*) | NY21-*frt* Δ*priC::frt-kan* | This work | NY21-*priC* | NY21-frt x P1 KRC002 |
| Strain, strain background (*Escherichia coli*) | NY20L-*frt* Δ*priC::frt-kan* | This work | NY20L-*priC* | NY20L-frt x P1 KRC002 |
| Strain, strain background (*Escherichia coli*) | NY20ATL-*frt* Δ*priC::frt-kan* | This work | NY20ATL-*priC* | NY20ATL-frt x P1 KRC002 |
| Strain, strain background (*Escherichia coli*) | MIT125 Δ*rnhA::cat* | This work | MIT125c | MIT125 x P1 MIT162 |
| Strain, strain background (*Escherichia coli*) | KRC004 Δ*rnhA::cat* | This work | KRC004c | KRC004 x P1 MIT162 |
| Antibody | anti-DnaA (Rabbit polyclonal) | *Kawakami et al., 2005* | | Immunoblot (1:3000) |
| Antibody | Goat Anti-Rabbit IgG (H+L)-AP Conjugate | Bio-Rad | Cat. #1706518 | Immunoblot (1:3000) |
| Recombinant DNA reagent | pBR*priC* (Plasmid) | This work | | pBR322 encoding *priC* |
| Recombinant DNA reagent | pING1 (Plasmid) | *Johnston et al., 1985* | | Vector bearing arabinose-inducible promoter |
| Recombinant DNA reagent | pKA234 (Plasmid) | *Kubota et al., 1997* | | pING1 encoding *dnaA* |
| Recombinant DNA reagent | pH136A (Plasmid) | *Sakiyama et al., 2018* | | pKA234 *dnaA* H136A |
| Recombinant DNA reagent | pFH (Plasmid) | This study | | pH136A mutated by Quick Change site-directed mutagenesis by primers F46A 1/2 |
| Recombinant DNA reagent | pKW44-1 (Plasmid) | *Ozaki et al., 2008* | | pKA234 *dnaA* V211A |
| Recombinant DNA reagent | pNA135 (Plasmid) | *Ishida et al., 2004* | | pBR322 bearing *diaA* gene |
| Recombinant DNA reagent | pBS*oriC* (Plasmid) | *Kawakami et al., 2005* | | pBluescript bearing a 678 bp chromosome-derived region including *oriC* |

*Appendix 1 Continued on next page*

*Appendix 1 Continued*

| Reagent type (species) or resource | Designation | Source or reference | Identifiers | Additional information |
|---|---|---|---|---|
| Recombinant DNA reagent | pET22b(+)-priC (Plasmid) | *Aramaki et al., 2015* | | pET22b(+) bearing *priC* under the T7 promoter |
| Recombinant DNA reagent | pTKM601 (Plasmid) | *Keyamura et al., 2007* | | pBAD/HisB bearing *diaA* |
| Sequence-based reagent | F46A 1 | *Keyamura et al., 2009* | PCR primer | GTACGCGCCAAACCGCGCGGTCCTTCGATTG GGTACG |
| Sequence-based reagent | F46A 2 | *Keyamura et al., 2009* | PCR primer | CGTACCCAATCGAAGGACCGCGCGGTTTGGCGCGTAC |
| Sequence-based reagent | ORI_1 | *Kasho et al., 2014* | qPCR primer | CTGTGAATGATCGGTGATC |
| Sequence-based reagent | KW*oriC*Rev | *Kasho et al., 2014* | qPCR primer | GTGGATAACTCTGTCAGGAAGCTTG |
| Sequence-based reagent | qoriK fw | *Brochu et al., 2018* | qPCR primer | CGAGACTTCAGCGACAGTTAAG |
| Sequence-based reagent | qoriK rev | *Brochu et al., 2018* | qPCR primer | CCTGCGGATATTTGCGATACA |
| Sequence-based reagent | STM419 | This work | qPCR primer | CGGACACCTTGTCTGACCTAAG |
| Sequence-based reagent | STM420 | This work | qPCR primer | AGTGTGAAAATGACCCTCTTGC |
| Commercial assay or kit | AP Conjugate Substrate Kit | Bio-Rad | Cat #1706432 | |

