## [Editor Report · eLife Assessment]

This manuscript reports findings of **fundamental** significance on how bacteria might load helicase for DNA replication when normal DnaA-based loading pathway is defective. It provides **convincing** genetic and biochemical evidence that helicase loading at the *E. coli oriC* is not (as previously assumed) exclusively performed by the DnaA initiator protein but can also be executed by PriC (whether this occurs specifically at oriC has not been addressed in vivo). This is a significant step forward in our understanding of bacterial replication initiation.

---

## [Referee Report · Reviewer #1 (Public review)]

Summary:

This manuscript reports the investigation of PriC activity during DNA replication initiation in *Escherichia coli*. It is reported that PriC is necessary for growth and control of DNA replication initiation under diverse conditions where helicase loading is perturbed at the chromosome origin oriC. A model is proposed where PriC loads helicase onto ssDNA at the open complex formed by DnaA at oriC. Reconstituted helicase loading assays in vitro are consistent with the model.

Strengths:

The complementary combination of genetics in vivo and reconstituted assays in vitro provide solid evidence to support the role of PriC at a replication origin.

The manuscript is well written and has a logical narrative.

The data provide new insight to how bacteria might load helicase at the replication origin when the wild-type DnaA-dependent loading pathway is perturbed.

Weakness:

It has not yet been established whether PriC localises at oriC in vivo under the conditions tested.

---

## [Referee Report · Reviewer #2 (Public review)]

This is a great paper. Yoshida et al. convincingly show that DnaA does not exclusively do loading of the replicative helicase at the *E. coli* oriC, but that PriC can also perform this function. Importantly, PriC seems to contribute to helicase loading even in wt cells albeit to a much lesser degree than DnaA. On the other hand, PriC takes a larger role in helicase loading during aberrant initiation, i.e. when the origin sequence is truncated or when the properties of initiation proteins are suboptimal. Here highlighted by mutations in dnaA or dnaC.

This a major finding because it clearly demonstrates that the two roles of DnaA in the initiation process can be separated into initially forming an open complex at the DUE region by binding/nucleation onto DnaA-boxes and second in loading of the helicase. Whereas these two functions are normally assumed to be coupled, the present data clearly show that they can be separated and that PriC can perform at least part of the helicase loading provided that an area of duplex opening is formed by DnaA.

This puts into questions the interpretation of a large body of previous work on mutagenesis of oriC and dnaA to find a minimal oriC/DnaA complex in many bacteria. In other words, mutants in which oriC is truncated/mutated may support initiation of replication and cell viability only in the presence of PriC. Such mutants are capable to generate single strand opening but may fail to load the helicase in absence of PriC. Similarly, dnaA mutants may generate aberrant complex on oriC that trigger strand opening but are incapable of loading DnaB unless PriC is present.

In the present work, the sequence of experiments presented is logical and the manuscript is clearly written and easy to follow. The very last part regarding PriC in cSDR replication does not add much to the story and may be omitted.

I have a few specific questions/comments

The partial complementation of the dnaC2 strain by PriC seems quite straightforward since this particular mutation leads to initiation arrest at the open complex stage and this sets the stage for PriC to load the helicase. The situation is somewhat different for dnaA46. Why is this mutation partly complemented by PriC at 37C? DnaA46 binds neither ATP nor ADP, yet it functions in initiation at permissive temperature. At nonpermssive temperature, it binds oriC as well but does not lead to initiation. Does the present data imply that the true initiation defect of DnaA46 lies in helicase loading? The authors need to comment on this in the text.

Relating to the above. In Fig. 3 it is shown that the pFH plasmid partly complement dnaA46 in a PriC dependent manner. Again, it would be nice to know the nature of the DnaA46 protein defect. It would be interesting to see how a pING1-dnaA46 plasmid performs in the experiment presented in Fig. 3.

---

## [Referee Report · Reviewer #3 (Public review)]

Summary:

At the abandoned replication fork, loading of DnaB helicase requires assistance from PriABC, repA, and other protein partners, but it does not require replication initiator protein, DnaA. In contrast, nucleotide-dependent DnaA binding at the specific functional elements is fundamental for helicase loading, leading to the DUE region's opening. However, the authors questioned in this study that in case of impeding replication at the bacterial chromosomal origins, oriC, a strategy similar to an abandoned replication fork for loading DnaB via bypassing the DnaA interaction step could be functional. The study by Yoshida et al. suggests that PriC could promote DnaB helicase loading on the chromosomal oriC ssDNA without interacting with the DnaA protein. The conclusions drawn supported by the evidence provided are compelling.

Strengths:

Understanding the mechanism of how DNA replication restarts via reloading the replisomes onto abandoned DNA replication forks is crucial. Notably, this knowledge becomes crucial to understanding how bacterial cells maintain DNA replication from a stalled replication fork when challenging or non-permissive conditions prevail. This critical study combines experiments to address a fundamental question of how DnaB helicase loading could occur when replication initiation impedes at the chromosomal origin, leading to replication restart.

---

## [Author Response]

The following is the authors’ response to the original reviews

**Public Reviews:**

**Reviewer #1 (Public review):**
Summary:This manuscript reports the investigation of PriC activity during DNA replication initiation in *Escherichia coli*. It is reported that PriC is necessary for the growth and control of DNA replication initiation under diverse conditions where helicase loading is perturbed at the chromosome origin oriC. A model is proposed where PriC loads helicase onto ssDNA at the open complex formed by DnaA at oriC. Reconstituted helicase loading assays in vitro support the model. The manuscript is well-written and has a logical narrative.

Thank you for understanding this study.

Major Questions/Comments:An important observation here is that a ΔpriC mutant alone displays under-replication, suggesting that this helicase loading pathway is physiologically relevant. Has this PriC phenotype been reported previously? If not, would it be possible to confirm this result using an independent experimental approach (e.g. marker frequency analysis or fluorescent reporter-operator systems)?

We thank Reviewer 1 for this comment. This study provides the first direct evidence for PriC’s role in initiation of chromosome replication. Given the change of the *oriC* copy number of ∆*priC* cells in non-stressed conditions is only slight, resolution of the suggested methods is clearly not high enough to distinguish the differences in the *oriC* copy number between *priC+* (WT) and ∆*priC* cells. Thus, to corroborate the ∆*priC* phenotype, we additionally analyzed using flow cytometry *priC+* and ∆*priC* cells growing under various nutrition and thermal conditions.

As shown in Figure 2-figure supplement 1 of the revised version, the fraction of cells with non-2^n^
*oriC* copies was slightly higher in ∆*priC* cells compared to *priC+* cells. Furthermore, when grown in M9 minimal medium at 37°C, ∆*priC* mutant cells exhibited slightly reduced ori/mass values. These are supportive to the idea that inhibition of replication initiation occurs at low frequency even in the WT *dnaA* and *dnaC* background, and that PriC function is necessary to ensure normal replication initiation. Related descriptions have been revised accordingly.

Is PriA necessary for the observed PriC activity at oriC? Is there evidence that PriC functions independently of PriA in vivo?

As described in Introduction of the original manuscript, PriA is a 3’-to-5’ helicase which specifically binds to the forked DNA with the 3’-end of the nascent DNA strand. Thus, structural specificity of target DNA is essentially different between PriA and PriC. Consistent with this, our in vitro data indicate that PriC alone is sufficient to rescue the abortive helicase loading at *oriC* (Figure 7), indicating that PriA is principally unnecessary for PriC activity at *oriC*. Consistently, as described in Introduction, PriC can interact with ssDNA to reload DnaB (Figure 1E). Nevertheless, a possibility that PriA might participate in the PriC-dependent DnaB loading rescue at *oriC* in vivo can not be completely excluded. However, elucidation of this possibility is clearly beyond the scope of the present study and should be analyzed in the future. An additional explanation has been included in Discussion of the revised version.

Is PriC helicase loading activity in vivo at the origin direct (the genetic analysis leaves other possibilities tenable)? Could PriC enrichment at oriC be detected using chromatin immunoprecipitation?

These are advanced questions about genomic dynamics of PriC. Given that PriC facilitates DnaB reloading at stalled replication forks (Figure 1E) (Heller and Marians, Mol Cell., 2005; Wessel et al., J Biol Chem, 2013; Wessel et al., J Biol Chem, 2016), PriC might interact with the whole genome and its localization might not necessarily exhibit a preference for *oriC* in growing cells. Analysis about these advanced questions is interesting but is beyond the scope of the present study and should be analyzed in the future study.

**Reviewer #2 (Public review):**
This is a great paper. Yoshida et al. convincingly show that DnaA does not exclusively do loading of the replicative helicase at the *E. coli* oriC, but that PriC can also perform this function. Importantly, PriC seems to contribute to helicase loading even in wt cells albeit to a much lesser degree than DnaA. On the other hand, PriC takes a larger role in helicase loading during aberrant initiation, i.e. when the origin sequence is truncated or when the properties of initiation proteins are suboptimal. Here highlighted by mutations in dnaA or dnaC.This is a major finding because it clearly demonstrates that the two roles of DnaA in the initiation process can be separated into initially forming an open complex at the DUE region by binding/nucleation onto DnaA-boxes and second by loading of the helicase. Whereas these two functions are normally assumed to be coupled, the present data clearly show that they can be separated and that PriC can perform at least part of the helicase loading provided that an area of duplex opening is formed by DnaA. This puts into question the interpretation of a large body of previous work on mutagenesis of oriC and dnaA to find a minimal oriC/DnaA complex in many bacteria. In other words, mutants in which oriC is truncated/mutated may support the initiation of replication and cell viability only in the presence of PriC. Such mutants are capable of generating single-strand openings but may fail to load the helicase in the absence of PriC. Similarly, dnaA mutants may generate an aberrant complex on oriC that trigger strand opening but are incapable of loading DnaB unless PriC is present.

We would like to thank Revierwer#2 for the very positive comments about our work.

In the present work, the sequence of experiments presented is logical and the manuscript is clearly written and easy to follow. The very last part regarding PriC in cSDR replication does not add much to the story and may be omitted.

Given that the role PriC in stimulating cSDR was unclear, we believe that our finding that PriC has little or no role in cSDR, despite being a negative result, is valuable for the general readership of eLife. To further assess impact of PriC on cSDR and as recommended by Referee #1, we carried out the chromosome loci copy-number analysis by the whole-genome sequencing. As shown in Figure 8-supplement 1 of the revised version, the results support our conclusion from the original version.

**Reviewer #3 (Public review):**
Summary:At the abandoned replication fork, loading of DnaB helicase requires assistance from PriABC, repA, and other protein partners, but it does not require replication initiator protein, DnaA. In contrast, nucleotide-dependent DnaA binding at the specific functional elements is fundamental for helicase loading, leading to the DUE region's opening. However, the authors questioned in this study that in case of impeding replication at the bacterial chromosomal origins, oriC, a strategy similar to an abandoned replication fork for loading DnaB via bypassing the DnaA interaction step could be functional. The study by Yoshida et al. suggests that PriC could promote DnaB helicase loading on the chromosomal oriC ssDNA without interacting with the DnaA protein. However, the conclusions drawn from the primarily qualitative data presented in the study could be slightly overwhelming and need supportive evidence.

Thank you for your understanding and careful comments.

Strengths:Understanding the mechanism of how DNA replication restarts via reloading the replisomes onto abandoned DNA replication forks is crucial. Notably, this knowledge becomes crucial to understanding how bacterial cells maintain DNA replication from a stalled replication fork when challenging or non-permissive conditions prevail. This critical study combines experiments to address a fundamental question of how DnaB helicase loading could occur when replication initiation impedes at the chromosomal origin, leading to replication restart.

Thank you for your understanding.

Weaknesses:The term colony formation used for a spotting assay could be misleading for apparent reasons. Both assess cell viability and growth; while colony formation is quantitative, spotting is qualitative. Particularly in this study, where differences appear minor but draw significant conclusions, the colony formation assays representing growth versus moderate or severe inhibition are a more precise measure of viability.

We used serial dilutions of the cell culture for the spotting assay and thus this assay should be referred as semi-quantitative rather than simply qualitative. For more quantitative assessment of viability, we analyzed the growth rates of cells and the chromosome replication activity using flow cytometry.

Figure 2The reduced number of two oriC copies per cell in the dnaA46priC-deficient strain was considered moderate inhibition. When combined with the data suggested by the dnaAC2priC-deficient strain containing two origins in cells with or without PriC (indicating no inhibition)-the conclusion was drawn that PriC rescue blocked replication via assisting DnaC-dependent DnaB loading step at oriC ssDNA.The results provided by Saifi B, Ferat JL. PLoS One. 2012;7(3):e33613 suggests the idea that in an asynchronous DnaA46 ts culture, the rate by which dividing cells start accumulating arrested replication forks might differ (indicated by the two subpopulations, one with single oriC and the other with two oriC). DnaA46 protein has significantly reduced ATP binding at 42C, and growing the strain at 42C for 40-80 minutes before releasing them at 30 C for 5 minutes has the probability that the two subpopulations may have differences in the active ATP-DnaA. The above could be why only 50% of cells contain two oriC. Releasing cells for more time before adding rifampicin and cephalexin could increase the number of cells with two oriCs. In contrast, DnaC2 cells have inactive helicase loader at 42 C but intact DnaA-ATP population (WT-DnaA at 42 or 30 C should not differ in ATP-binding). Once released at 30 C, the reduced but active DnaC population could assist in loading DnaB to DnaA, engaged in normal replication initiation, and thus should appear with two oriC in a PriC-independent manner.

This is a question about *dnaA46* Δ*priC* mutant cells. Inhibition of the replication forks causes inhibition of RIDA (the DNA-clamp complex-dependent DnaA-ATP hydrolysis) system, resulting in the increase of ATP-DnaA molecules (Kurokawa et al. (1999) EMBO J.). Thus, if Δ*priC* inhibits the replication forks significantly, the ATP-DnaA level should increase and initiation should be stimulated. However, the results of Figure 2BC are opposite, indicating inhibition of initiation by Δ*priC*. Thus, we infer that the inhibition of initiation in the Δ*priC* cells is not related to possible changes in the ATP-DnaA level. Even if the ATP-DnaA levels are different in subpopulations in *dnaA46* cells, Δ*priC* mutation should not affect the ATP-DnaA levels significantly. Thus, we infer that even in *dnaA46* Δ*priC* mutant cells, Δ*priC* mutation directly affect initiation mechanisms, rather than indirectly through the ATP-DnaA levels.

Broadly, the evidence provided by the authors may support the primary hypothesis. Still, it could call for an alternative hypothesis: PriC involvement in stabilizing the DnaA-DnaB complex (this possibility could exist here). To prove that the conclusions made from the set of experiments in Figures 2 and 3, which laid the foundations for supporting the primary hypothesis, require insights using on/off rates of DnaB loading onto DnaA and the stability of the complexes in the presence or absence of PriC, I have a few other reasons to consider the latter arguments.

This is a very careful consideration. However, we infer that stabilization of the DnaA-DnaB interaction by PriC, even if present, does not always result in stimulation of DnaB loading to *oriC.* Given that interactions between DnaA and DnaB during DnaB loading to *oriC* are highly dynamic and complicated with multiple steps, stabilization of the DnaA-DnaB interaction by PriC, even if it occurs, has a considerable risk of inhibiting the DnaB loading by constructing abortive complexes. In addition, DnaA-DiaA binding is very tight and stable (Keyamura et al., 2007, 2009). Even if WT DnaA and WT DnaB are present, PriC can rescue the initiation defects of *oriC* mutants. Based on these facts and the known characteristics of PriC as explained in Introduction, it is more reasonable to infer that PriC provides a bypass of DnaB loading even at *oriC*, as proposed for the mechanism at the stalled replication fork. However, we cannot completely rule out the indicated possibility and these explanations are included in the revised version.

Figure 3One should consider the fact that dnA46 is present in these cells. Overexpressing pdnaAFH could produce mixed multimers containing subunits of DnaA46 (reduced ATP binding) and DnaAFH (reduced DnaB binding). Both have intact DnaA-DnaA oligomerization ability. The cooperativity between the two functions by a subpopulation of two DnaA variants may compensate for the individual deficiencies, making a population of an active protein, which in the presence of PriC could lead to the promotion of the stable DnaA: DnaBC complexes, able to initiate replication. In the light of results presented in Hayashi et al. and J Biol Chem. 2020 Aug 7;295(32):11131-11143, where mutant DnaBL160A identified was shown to be impaired in DnaA binding but contained an active helicase function and still inhibited for growth; how one could explain the hypothesis presented in this manuscript. If PriC-assisted helicase loading could bypass DnaA interaction, then how growth inhibition in a strain carrying DnaBL160A should be described. However, seeing the results in light of the alternative possibility that PriC assists in stabilizing the DnaA: DnaBC complex is more compatible with the previously published data.

Unfortunately, in this comment, there is a crucial misunderstanding in the growth of cells bearing DnaA L160A. Hayashi et al. reported that the *dnaB*(Ts) cells bearing the *dnaB* L160A allele grew slowly and formed colonies even at 42°C. This feature is similar to the growth of *dnaA46* cells bearing *dnaA* F46A H136A allele (Figure 2). Thus, the results of *dnaB* L160A cells are consistent with our model and support the idea that PriC partially rescues the growth inhibition of cells bearing the DnaB L160A allele by bypassing the strict requirement for the DnaA-DnaB interaction. Nevertheless, we have to be careful about a possibility that DnaB L160A could affect interaction with PriC, which we are going to investigate for a future paper.

As suggested, if mixed complexes of DnaA46 and DnaA F46A H136A proteins are formed, those might retain partial activities in *oriC* unwinding and DnaB interaction although those cells are inviable at 42°C without PriC. It is noteworthy that in the specific *oriC* mutants which are impaired in DnaB loading (e.g., Left-*oriC*), PriC effectively rescues the initiation and cell growth. In these cells, both DnaA and DnaB are intact. Thus, the idea that only mutant DnaA (or DnaB) protein is simulated specifically via PriC interaction is invalid. Even in cells bearing wild-type *oriC*, DnaA and DnaB, contribution of PriC for initiation is detected.

In addition, as described in the above response, given that interactions between DnaA and DnaB during DnaB loading to *oriC* are very dynamic and complicated with multiple steps, stabilization of the DnaA-DnaB interaction by PriC, even if present, would not simply result in stimulation of DnaB loading to *oriC*; rather we think a probability that it would inhibit the DnaB loading by constructing abortive complexes. Based on the known characteristics of PriC as explained in Introduction, it is more reasonable to infer that PriC provides a bypass of DnaB loading even at *oriC*, as proposed for the mechanism at the stalled replication fork.

However, we cannot completely rule out the indicated possibility and this explanation has been described in the revised version as noted in response to the above question.

Figure 4Overexpression of DiaA could contribute to removing a higher number of DnaA populations. This could be more aggravated in the absence of PriC (DiaA could titrate out more DnaA)-the complex formed between DnaA: DnaBC is not stable, therefore reduced DUE opening and replication initiation leading to growth inhibition (Fig. 4A ∆priC-pNA135). Figure 7C: Again, in the absence of PriC, the reduced stability of DnaA: DnaBC complex leaves more DnaA to titrate out by DiaA, and thus less Form I*. However, adding PriC stabilizes the DnaA: DnaBC hetero-complexes, with reduced DnaA titration by DiaA, producing additional Form I*. Adding a panel with DnaBL160A that does not interact with DnaA but contains helicase activity could be helpful. Would the inclusion of PriC increase the ability of mutant helicase to produce additional Form I*?

Unfortunately, the proposed idea is biased disregarding the fact that DiaA effectively stimulates assembling processes of DnaA molecules at *oriC*. As *oriC* contains multiple DnaA boxes and multiple DnaA molecules are recruited there, DiaA will efficiently facilitate assembling of DnaA molecules on *oriC*. Even DnaA molecules of DnaA-DiaA complexes can efficiently bind to *oriC*. This is consistent with in vitro experiments showing that higher levels of DiaA stimulate assembly of DnaA molecules and *oriC* unwinding (i.e., DUE opening) but even excessive levels of DiaA do not inhibit those reactions (Keyamura et al., J. Biol. Chem. (2009) 284, 25038-25050). However, as shown in Figure 9, DiaA tightly binds to the specific site of DnaA which is the same as the DnaB L160-binding site, which causes inhibition of DnaA-DnaB binding (*ibid*). These are consistent with in vivo experiments, and concordantly consistent with the idea that the excessive DiaA level inhibits interaction and loading of DnaB by the DnaA-*oriC* complexes, but not *oriC* unwinding (i.e., DUE opening) in vivo. Also, as mentioned above, we do not consider that stabilization of DnaA-DnaBC complex simply results in stimulation of DnaB loading to *oriC*. Based on the known characteristics of PriC, it is more reasonable to infer that PriC provides a bypass of DnaB loading even at *oriC*, as proposed for the mechanism at the stalled replication fork (Figure 1E), as described in the above response.

As for DnaB L160A, as mentioned above, we are currently investigating interaction modes between DnaB and PriC. While investigating DnaB L160A could further support our model, we believe its contribution to the present manuscript would be incremental. In addition, there is a possibility that DnaA L160A could affect interaction with PriC. Thus, analysis of DnaB mutants in this PriC rescue mechanisms should be addressed in future study.

Figure 5The interpretation is that colony formation of the Left-oriC ∆priC double mutant was markedly compromised at 37°C (Figure 5B), and 256 the growth defects of the Left-oriC mutant at 25{degree sign}C and 30{degree sign}C were aggravated. However, prima facia, the relative differences in the growth of cells containing and lacking PriC are similar. Quantitative colony-forming data is required to claim these results. Otherwise, it is slightly confusing.

The indicated concern was raised due to our typing error lacking *∆priC*. In the revised manuscript, we have amended as follows: the cell growth of the Left-*oriC* ∆*priC* double mutant was markedly compromised at 37°C and moderately reduced at 25°C and 30°C (Figure 5B).

A minor suggestion is to include cells expressing PriC using plasmid DNA to show that adding PriC should reverse the growth defect of dnaA46 and dnaC2 strains at non-permissive temperatures. The same should be added at other appropriate places.

Even in the presence of PriC, unwinding of *oriC* and DnaB helicase loading to the wound *oriC* require DnaA and DnaC activities as indicated by previous studies (see for a review, Windgassen et al., (2018) Nucleic Acids Res. 46, 504-519). Thus, *dnaA46* cells and *dnaC2* cells bearing pBR322-*priC* can not grow at 42°C and 37°C (as follows). These are reasonable results. However, at semi-permissive temperatures (37°C for *dnaA46* and 35°C for *dnaC2*), slight stimulation of the cell growth by pBR322-*priC* might be barely observed (Figure 2-supplement 1 of the revised version). These suggest that the intrinsic level of PriC is functionally nearly sufficient. This explanation has been included in the revised version.

**Author response image 1. sa4fig1:** 

**Recommendations for the authors:**

**Reviewer #1 (Recommendations for the authors):**
Line 38. "in assembly of the replisome".

Corrected.

Line 137. "specifically" rather than specificity.

Corrected.

Line 139. "at" rather than by.

Corrected.

The DnaA46 protein variant contains two amino acid substitutions (A184V and H252Y) within the AAA+ motif. H136 appears to reside adjacent to A184 in structure. Is A184V mutation causative?

The DnaA H136A and A184V alleles are responsible for different defects. Indeed, the DnaA A184V variant is thermolabile and defective in ATP binding whereas the H136A variant retains ATP binding but impairs DnaB loading (Carr and Kaguni, Mol. Microbiol., 1996; Sakiyama et al., Front. Microbiol., 2018). These observations strongly support the view that the phenotype of the DnaA H136A allele is independent of that of the DnaA A184V allele.

Figure 2A. Regarding the dnaA46 allele grown at 37°C.Individual colonies cannot be resolved. Is an image from a later time-point available?

We have replaced the original image with one from another replicate that provides better resolution. Please see Figure 2A in the revised version.

Figure 2C. Quantification of the number of cells with more than one chromosome equivalent in the dnaC2 ΔpriC strain. The plot from flow cytometry appears to show >20% of cells with only 1 genome. Are these numbers correct?

Thank you for this careful comment. We quantified the peaks more strictly, but the percentages were noy largely changed. To improve resolution of the DNA profiles, we have changed the range of the x-axis in panels B and C of Figure 2 in the revised version.

Figure 3. Are both F46A and H136A mutations in the plasmid-encoded dnaA necessary?

Yes. The related explanation is included in the Discussion section (the third paragraph) of the original manuscript. As described there, *dnaA46* cells expressing the DnaA H136A single mutant exhibited severe defects in cell growth even in the presence of PriC (Sakiyama et al., 2018). The His136 residue is located within the weak, secondary DnaB interaction region in DnaA, and is crucial for DnaB loading onto *oriC* ssDNA. Given domain I in DnaA H136A can stably tether DnaB-DnaC complexes to DnaA complexes on *oriC* (Sakiyama et al., 2018), we infer that *oriC*-DnaA complexes including DnaA H136A stably bind DnaB via DnaA domain I as an abortive complex, which inhibits functional interaction between PriC and DnaB as well as DnaB loading to *oriC* DNA.

As for DnaA F46A mutant, our previous studies show that DnaA F46A has a limited residual activity in vivo (unlike in vitro), and allows slow growth of cells. As the stable DnaA-DnaB binding is partially impaired in vivo in DnaA F46A, this feature is consistent with the above ideas. Thus, both F46A and H136A mutations are required for severer inhibition of DnaB loading. This is additionally described in the revised Discussion.

Figure 3. Is the DnaA variant carrying F46A and H136A substitutions stably expressed in vivo?

We have performed western blotting, demonstrating that the DnaA variant carrying F46A and H136A substitutions is stable in vivo. In the revised version, we have added new data to Figure 3-figure supplement 1 and relevant description to the main text as follows:

Western blotting demonstrated that the expression levels were comparable between WT DnaA and DnaA F46A H136A double mutant (Figure 3-figure supplement 1).

Figure 5A. Should the dashed line extending down from I2 reach the R4Tma construct?

We have amended the indicated line appropriately.

Figure 6C. It was surprising that the strain combining the subATL mutant with ΔpriC displayed a pronounced under-initiation profile by flow cytometry, and yet there was no growth defect observed (see Figure 6B). This seems to contrast with results using the R4Tma origin, where the ΔpriC mutant produced a relatively modest change to the flow cytometry profile, and yet growth was perturbed (Figure 5C-D). How might these observations be interpreted? Is the absolute frequency of DNA replication initiation critical?

Please note that, in *E. coli*, initiation activity corelates closely with the numbers of *oriC* copies per cell mass (ori/mass), rather than the apparent DNA profiles measured by flow cytometer. When cells were grown in LB at 30°C, the mean ori/mass values were as follows: 0.34 for R4*Tma* priC, 0.51 for R4*Tma*, 0.82 for DATL priC, 0.99 for DATL (Figures 5 & 6 in the original manuscript). These values closely correspond to the cell growth ability shown in Figure 5C in the original manuscript.

In the revised manuscript, we have cited appropriate references for introduction of the ori/mass values as follows.

To estimate the number of *oriC* copies per unit cell mass (ori/mass) as a proxy for initiation activity (Sakiyama et al., 2017, 2022),

Line 295. Reference for Form I* assay should cite the original publication.

Done. The following paper is additionally cited.

Baker, T. A., Sekimizu, K., Funnell, B. E., and Kornberg, A. (1986). Extensive unwinding of the plasmid template during staged enzymatic initiation of DNA replication from the origin of the *Escherichia coli* chromosome. Cell 45, 53–64.doi: 10.1016/0092-8674(86)90537-4

**Reviewer #2 (Recommendations for the authors):**
The partial complementation of the dnaC2 strain by PriC seems quite straightforward since this particular mutation leads to initiation arrest at the open complex stage and this sets the stage for PriC to load the helicase. The situation is somewhat different for dnaA46. Why is this mutation partly complemented by PriC at 37C? DnaA46 binds neither ATP nor ADP, yet it functions in initiation at permissive temperature. At nonpermissive temperature, it binds oriC as well but does not lead to initiation. Does the present data imply that the true initiation defect of DnaA46 lies in helicase loading? The authors need to comment on this in the text.

Given the thermolabile propensity of the DnaA46 protein, it is presumable that DnaA46 protein becomes partially denatured at the sub-permissive temperature of 37°C. This partial denaturation should impair both origin unwinding and helicase loading, though not to the extent that cell viability is lost. The *priC* deletion should further exacerbate helicase loading defects by inhibiting the bypass mechanism, resulting in the lethality of *dnaA46* cells at this temperature. This explanation is included in the revised Discussion section.

Relating to the above. In Figure 3 it is shown that the pFH plasmid partly complements dnaA46 in a PriC-dependent manner. Again, it would be nice to know the nature of the DnaA46 protein defect. It would be interesting to see how a pING1-dnaA46 plasmid performs in the experiment presented in Figure 3.

A previous paper showed that multicopy supply of DnaA46 can suppress temperature sensitivity of the *dnaA46* cells (Rao and Kuzminov, G3, 2022). This is reasonable in that DnaA46 has a rapid degradation rate unlike wild-type DnaA. As DnaA46 preserves the intact sequences in DnaB binding sites such as G21, F46 and H136, the suppression would not depend on PriC but would be due to the dosage effect.

Figure 8 B: The authors should either remove the data or show a genome coverage: it is not clear that yapB is a good reference. A genome coverage would be nice, and show whether initiation can occur at oriC even if it is not the major place of initiation in a rnhA mutant.

As suggested, we carried out the chromosome loci copy-number analysis by whole-genome sequencing to assess impact of PriC on cSDR. The new data are shown in Figure 8-supplement 1 with relevant descriptions of the main text of the revised version as shown below. Briefly, results of the chromosome loci copy-number analysis are consistent with those of real-time qPCR (Figure 8B). Given that the role PriC in stimulating cSDR was unclear, we believe that our finding that PriC has little or no role in cSDR, despite being a negative result, is valuable for the general readership of eLife.

Line 38-39: .....resulting in replisome assembly.

Corrected.

Line 48: Something is wrong with the Michel reference. Also in the reference list.

Corrected

Line 156: replace retarded with reduced.

Corrected.

Line 171 and elsewhere: WT priC cells is somewhat misleading. Isn't this simply PriC+ cells?

Yes. We have revised the wording to “*priC+*” for clarity.

Line 349-350: "the oriC copy number ratio of the dnaA46 DpriC double mutant was lower than that of the dnaA46 single mutant....". This is only provided growth rate of the strains is the same.

These strains exhibited similar growth rates. This is included in the Result section of the revised manuscript as follows: At the permissive temperature, despite having similar growth rates, the *oriC* copy number ratio of the *dnaA46* ∆*priC* double mutant strain was lower than that of the *dnaA46* single mutant.

**Reviewer #3 (Recommendations for the authors):**
I would suggest improved or additional experiments, data, or analyses.

The revised version includes improved or additional experiments, data, or analyses.